# Towards improved retrieval of aerosol properties with the new Meteosat Third Generation-Imager geostationary satellite

Adèle Georgeot[1], Xavier Ceamanos[1], Jean-Luc Attié[2], Daniel Juncu[1], Josef Gasteiger[3], and Mathieu Compiègne[4]

[1]CNRM, Météo-France/CNRS/Université de Toulouse, Toulouse, France
[2]LAERO, OMP/Université de Toulouse/CNRS, Toulouse, France
[3]Hamtec Consulting GmbH at EUMETSAT, Darmstadt, Germany
[4]Hygeos, Lille, France

**Correspondence:** Xavier Ceamanos (xavier.ceamanos@meteo.fr)

**Abstract.**

Aerosols have significant effects on Earth, which vary according to the type of these atmospheric particles. Different observing systems exist today to monitor aerosols, mainly through the retrieval of aerosol optical depth (AOD), among which meteorological satellites in geostationary orbit provide unique information thanks to their acquisition of several Earth's images per hour. The third generation of European geostationary satellites, Meteosat Third Generation-Imager with the onboard Flexible Combined Imager (FCI) operational since December 2024, brings new possibilities for aerosol remote sensing compared to its predecessor, Meteosat Second Generation, with the Spinning Enhanced Visible and Infrared Imager (SEVIRI) on board. This article assesses the improvements in aerosol retrieval that will be made possible thanks to FCI, based on realistically generated synthetic data. Two case studies corresponding to challenging aerosol retrieval situations are simulated, a dust outbreak in North Africa and the wildfire season in South West Africa. First, synthetic data are used to study the potential for AOD retrieval of new FCI spectral channels in comparison to SEVIRI's. Results prove that channel VIS04 (centered at 444 nm) is the best suited for this task, with a significant decrease in retrieval error (root square mean error by 23% and mean bias error by 65%) in comparison to AOD estimated from the SEVIRI-heritage channel VIS06 (centered at 640 nm). Second, the FCI capabilities to further characterize aerosol particles are investigated, with the development of a method to simultaneously estimate AOD and fine mode fraction (FMF), which is linked to particle size distribution and therefore aerosol type. This is achieved by exploiting near-infrared channel NIR22 (centered at 2250 nm, and being sensitive to coarse particles only) in addition to channel VIS04 using an optimal estimation approach and considering the contributions of fine and coarse aerosol modes separately. Experiments show that, except under certain unfavorable conditions, the joint retrieval of AOD and FMF is possible, even when fast radiative transfer models adapted to operational processing are used. This article demonstrates the possibility of obtaining advanced high temporal frequency aerosol observations from FCI and opens pathways for the future study of aerosol diurnal variations from space.

## 1 Introduction

Aerosols are small particles suspended in the atmosphere that vary widely in composition and size, ranging from 1 nm to 10 μm in diameter. They have various direct and indirect effects on weather, climate, air quality, air transport and defense (Boucher, 2015). Aerosol direct effects on climate, for example, depend mostly on the particles' radiative properties, which determine their absorption and scattering of solar radiation. Radiative effects are controlled by the aerosols' optical properties, which in turn are related to chemical composition and particle size. For example, black carbon aerosols, composed of fine particles, have a warming effect on the planet, whereas desert dust aerosols, corresponding to coarse particles, have a cooling effect (Li et al. (2013), Gkikas et al. (2018)). Mallet et al. (2020) proved that aerosols' direct and semi-direct effects are sensitive to their absorbing properties and therefore depend on single scattering albedo (SSA), and Matsui et al. (2018) showed that, within the same aerosol type, particle size can modify radiative effects. Indirect effects such as cloud formation and precipitation efficiency are also related to particle size. More precisely, it has been proved important to know the number of particles above a given size to predict the indirect effects of aerosols on clouds (Mahowald et al., 2014). All these studies highlight the importance of improving aerosol properties characterization.

One characteristic we want to address in this study is the estimation of particle size distribution. One way to constrain this aerosol parameter is through the estimation of fine mode fraction (FMF), which represents the contribution of fine particles (approximately below 1 μm in diameter) to the total aerosol optical depth (AOD), with values between 0 and 1. Knowing FMF can help determining aerosols radiative forcing (Chung et al., 2016) and estimating PM2.5 (particulate matter with diameters below 2.5 μm) (Zhang et al., 2015). Another key feature of aerosols, which is still poorly understood today, is their rapid variation with time, such as in the occurrence of extreme events including dust outbreaks, intense wildfire emissions and volcanic eruptions (Plu et al., 2021). Furthermore, knowing the diurnal cycle of some aerosol species such as desert dust and pollution is important for weather forecasting and climate modeling (Kocha et al. (2013), Xu et al. (2016)), and can help better understand carbon monoxide variations and sources (Buchholz et al., 2021).

One tool to monitor these afore-mentioned aerosol characteristics is satellite data which offer the combination of covering large spatial scales and (in case of geostationary satellites) high frequency of observations. Indeed, geostationary Earth orbit satellites are able to observe the exact same Earth's region (i.e., the so-called geostationary disk, roughly covering one third of the planet) all day long thanks to their location at around 35,800 km above the surface. This constant coverage of the Earth makes it possible to monitor aerosols' diurnal variations at sub-hourly frequencies with large spatial coverage. Descheemaecker et al. (2019) proved the benefits of assimilating geostationary hourly aerosol data in an atmospheric forecasting model, outperforming the 1 or 2 measurements per day provided by low Earth orbit satellites. Similar conclusions were also drawn by Plu et al. (2021) when modeling the volcanic plume from the Eyjafjallajökull eruption in May 2010. In that study, the assimilation of MODIS (Moderate-Resolution Imaging Spectroradiometer) aerosol retrievals in the MOCAGE (Modèle de Chimie Atmosphérique de Grande Echelle) model did not improve the ash plume forecasting due to the poor temporal frequency of the MODIS satellite data.

Meteosat Second Generation (MSG) is a geostationary satellite operated by EUMETSAT (European Organisation for the Exploitation of Meteorological Satellites) located at 0° along the Equator, and therefore covering Europe, Africa and South America (see Fig. 1). The multi-spectral imaging radiometer SEVIRI (Spinning Enhanced Visible and Infrared Imager) on board MSG has an acquisition frequency of 15 minutes, which enables the sub-hourly retrieval of AOD (Luffarelli et al. (2019), Ceamanos et al. (2023)). Contrary to more recent geostationary imagers, SEVIRI has only three spectral channels in the visible and near infrared range, with the "red" channel VIS06 centered at 635 nm being the shortest wavelength available and the main aerosol information source. Nonetheless, channel VIS06 is not perfectly suited to aerosol properties retrieval, since land surfaces are generally bright at this wavelength. This makes it challenging to disentangle aerosol signal from top of atmosphere satellite measurements, especially when geometry is not favorable (Ceamanos et al., 2023). Shorter visible wavelengths are known to be generally better for aerosol remote sensing, as it can be seen in the well-known MODIS-based algorithms working with "blue" spectral channels (Hsu et al. (2013), Levy et al. (2013), Lyapustin et al. (2018)), where land surfaces are generally darker (Zoogman et al., 2016) and aerosols are brighter or equally bright as in the red wavelengths, therefore increasing the aerosol information content. As for FMF retrieval, existing algorithms use multiple channels in the visible and near infrared to exploit the different spectral signature of fine and coarse aerosols (Lyapustin et al. (2011), Choi et al. (2018), Zhang et al. (2021), Limbacher et al. (2024)). For example, the MAGARA (Multi-Angle Geostationary Aerosol Retrieval Algorithm) algorithm (Limbacher et al., 2024) uses data from several bands from GOES/ABI (Geostationary Operational Environmental Satellite/Advanced Baseline Imager) between 0.470 nm to 2.25 $\mu$m, whereas Lyapustin et al. (2011) uses MODIS spectral channels centered at 0.47, 0.67 and 2.1 $\mu$m.

The recent launch of the EUMETSAT next generation geostationary satellite, Meteosat Third Generation-Imager (MTG-I), is expected to enable a better aerosol characterization across the Meteosat disk (Descheemaecker et al. (2019), Aoun (2016)) thanks to its advanced multi-spectral imaging radiometer FCI (Flexible Combined Imager), operational since December 2024. In addition to the increase in acquisition frequency and spatial resolution of FCI with respect to SEVIRI (10 minutes and 1 km versus 15 minutes and 3 km) (Holmlund et al., 2021), FCI has 5 additional spectral channels in the visible and near infrared ranges as shown in Fig. 2. Some of these new channels are expected to enable a better characterization of aerosol properties. For example, the VIS04 "blue" and VIS05 "green" visible channels, centered at 444 nm and 510 nm respectively, are expected to be more sensitive to aerosols, in particular compared to the SEVIRI-heritage channel VIS06, due to the previously mentioned different spectral reflectance of land surfaces and aerosols (see Fig. 2). This can be seen in the magnified sub-figure in Fig. 1, which shows how smoke plumes emitted in September 2024 by wildfires in Northern Portugal are visible over a wider spatial extent in the FCI true color image due to the better sensitivity to aerosols of its visible channels (VIS04, VIS05, VIS06) compared to SEVIRI channels VIS06, VIS08 and NIR16. Moreover, FCI channel NIR22 centered at 2250 nm is expected to help distinguishing among aerosol species because only coarse particles such as desert dust scatter radiation at these wavelengths (Fig. 2).

In this article we evaluate the new capabilities offered by the FCI instrument for diurnal aerosol characterization at high temporal frequency, with a focus on AOD and FMF retrieval. In the first part, we quantitatively assess the possible increase in AOD retrieval accuracy with FCI in comparison to what is currently achieved with SEVIRI (e.g., Ceamanos et al., 2023). In

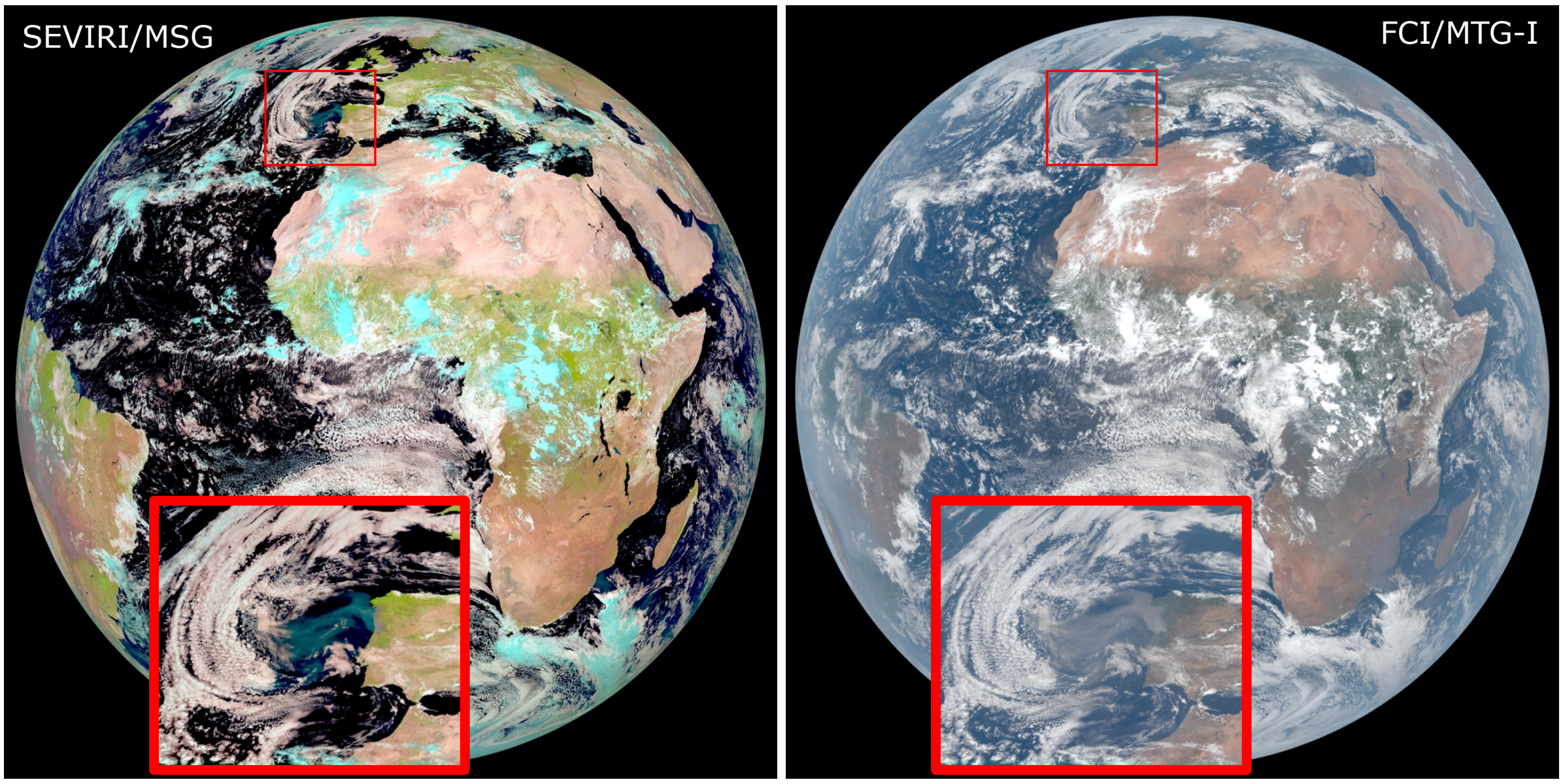

**Figure 1.** Full disk pictures of SEVIRI and FCI on 18 September 2024 at 12:00 UTC. The magnified sub-figure shows the plumes emitted by wildfires in the north of Portugal. The false color composite of SEVIRI is obtained by combining the radiance measured by spectral channels VIS06 (in blue), VIS08 (in green), and IR016 (in red), and the true color composite of FCI is obtained with radiance from VIS04 (in blue), VIS05 (in green) and VIS06 (in red). Note that FCI data were still pre-operational at these dates.

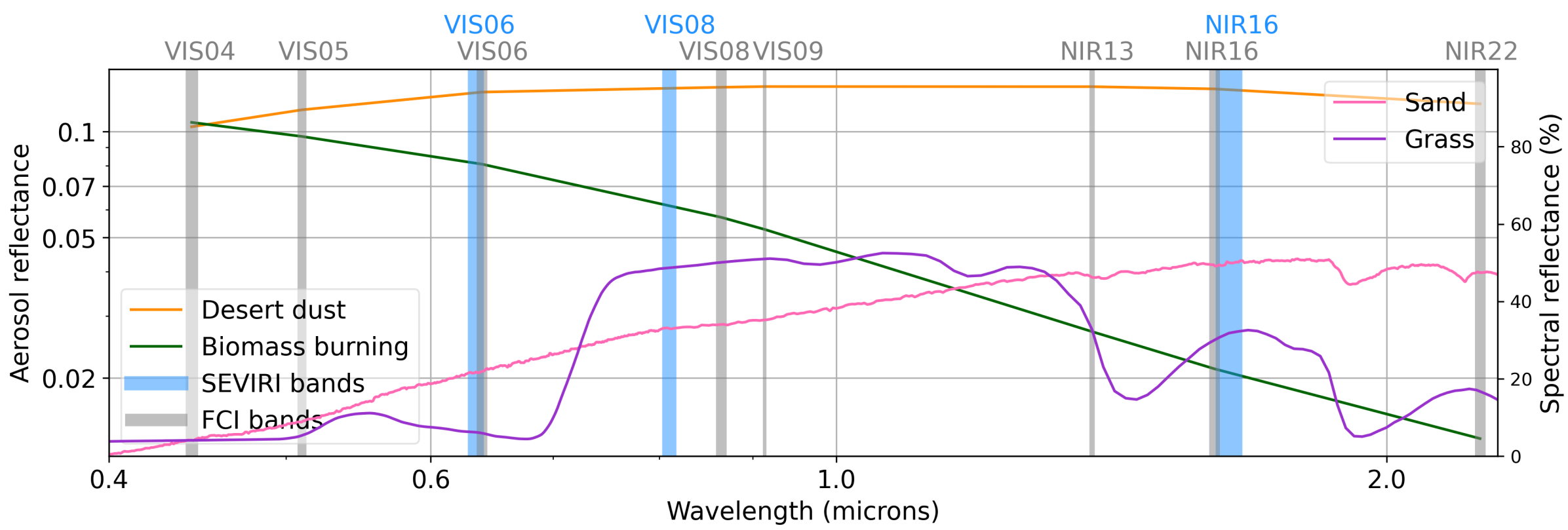

**Figure 2.** Visible and near infrared channels of FCI (in gray color) and SEVIRI (in blue color) imagers, and spectral reflectance corresponding to two different aerosol types and two land surface cover types. Channels central wavelength and width are from Holmlund et al. (2021) and aerosol reflectance was computed with the FCI simulator presented in this study. Spectral reflectances are from the ECOSTRESS spectral library version 1.0 (Meerdink et al., 2019).

particular, the benefits of using the new short visible channels from FCI, mainly VIS04, are evaluated through a comparison to SEVIRI-like VIS06 AOD retrievals. In the second part, we evaluate the possibilities to further characterize aerosol particles by performing a joint retrieval of AOD and FMF at high temporal frequency, using FCI spectral channels VIS04 and NIR22. Experiments are carried out on realistic FCI-like synthetic data that were generated with the goal of mimicking two real aerosol events challenging for remote sensing, a desert dust outbreak in North West Africa with the presence of bright surfaces, and the wildfire season in South West Africa corresponding to unfavorable satellite geometries. More precisely, we evaluate the FCI potential using optimal estimation-based methods developed in this study to (i) quantify the aerosol information content of these synthetic data and (ii) invert them for AOD and FMF. Furthermore, inversion experiments are carried out using fast radiative transfer models to assess the possibility of meeting the constraints of operational processing.

This article is organized as follows. First, we give an overview of the study introducing its main concepts, the selected aerosol events and the generation of the FCI-like synthetic data in Sect. 2. Second, we present the assessment of new FCI visible channels for AOD estimation, including the results of sensitivity studies and inversion experiments, in Sect. 3. Third, the joint retrieval of AOD and FMF is discussed in Sect. 4, presenting the inversion method and the obtained results. Finally, conclusions are drawn in Sect. 5.

## 2 Experiment Setup

### 2.1 Overview

The main objective of this study is to assess the performances of the new instrument FCI for aerosol retrieval, mainly in challenging situations such as dust outbreaks in desert areas. Several experiments are conducted to address this question using realistically-generated FCI-like synthetic data. We can separate our study into two main steps: the simulation of synthetic data and their processing for aerosol properties inversion.

The simulation of FCI-like synthetic data is carried out for two case studies based on real aerosols events that were monitored by ground- and space-based instruments (Sect. 2.2). The end product of this first step is a set of satellite reflectance time series, at the sub-hourly frequency and covering the full aerosol event, one for each of the sites considered in our study. Simulations are made with the accurate doubling-adding (DOAD) radiative transfer model (De Haan et al., 1987). All inputs required for simulation (e.g., aerosol properties, surface reflectance, sensor characteristics, solar/view angles) are obtained from realistic data (Sect. 2.3). Synthetic data are simulated without including atmospheric gases for the sake of simplicity and to focus on the retrieval of aerosol properties. This choice results in the simulation of top of aerosol layer (TOL) reflectance, resulting from the contributions from aerosols and the surface only.

The second step consists in retrieving AOD (and FMF in a later step) from the simulated FCI-like synthetic data. This data inversion needs some of the inputs used for simulation including surface reflectance and angles, and uses fast radiative transfer models, mostly the Modified Sobolev Approximation (MSA, Katsev et al. (2010), see details in Appendix B), to be compliant with the constraints of large data volume operational processing. Similar to Georgeot et al. (2024), the optimal estimation-

based Levenberg-Marquardt method (Sect. 2.4) is used for numerical inversion. Optimal estimation is also used to evaluate the information content enclosed in FCI-like synthetic data.

## 2.2 Case Studies

Synthetic data are simulated for two case studies corresponding to real aerosol events, one corresponding to a desert dust outbreak in North West Africa (CS1) and one corresponding to the wildfire season in South West Africa (CS2). These events were monitored by the AERONET stations (Holben et al., 1998) listed in Table 1, which also provides information on the selected case studies.

**Table 1.** Main characteristics of the two case studies considered in this study. $N$ is the total number of AERONET AOD measurements available in each case.

| Case Study | Aerosol Type | Dates | Region | AERONET Stations | $N$ |
|---|---|---|---|---|---|
| CS1 | Desert Dust | 18/06/2016 to 27/06/2016 | North West Africa | *Capo Verde*, *Dakar*, *Izana*, *La Laguna*, *Saada*, *Teide* | 2218 |
| CS2 | Biomass Burning | 13/09/2016 to 28/09/2016 | South West Africa | *Ascension Island*, *Lubango*, *Mongu Inn*, *Namibe* | 1460 |

The desert dust event selected for CS1 started on 18 June 2016 in the Sahara desert, and reached the Atlantic Ocean and the Canary Islands over a few days. To reproduce this event we select 10 days, from 18 to 27 June 2016, of AOD measured by 6 AERONET stations, *Dakar*, *Saada*, *Capo Verde*, *Izana*, *Teide* and *La Laguna*. The location of these stations enables to cover different regions affected by the dust event and to consider different Meteosat geometric configurations (Fig. 3). The satellite synoptic view shown in this figure corresponding to 24 June shows a massive dust plume leaving the Sahara desert and entering the Atlantic Ocean. CS1 corresponds to a challenging case for aerosol retrieval over land because of the generally high surface brightness in North West Africa.

The biomass burning event selected for CS2 started on 13 September 2016 and lasted two weeks. To simulate this event we select AOD measurements from 13 to 28 September 2016 from 4 AERONET stations, *Namibe*, *Lubango*, *Mongu Inn* and *Ascension Island*. Figure 3 shows the location of these ground stations, as well as a satellite synoptic view of the smoke plume (mainly visible over land, surrounding *Mongu Inn*). According to AERONET data (not shown here), the wildfire smoke plume emitted in South Africa reached Ascension Island 12 days after, on 25 September, after traveling around 3,000 km. Aerosol retrieval is challenging for CS2 because of the highly backscattering geometry of Meteosat observations over South Africa at the end of the southern winter, which results in low aerosol scattering and high surface reflectance.

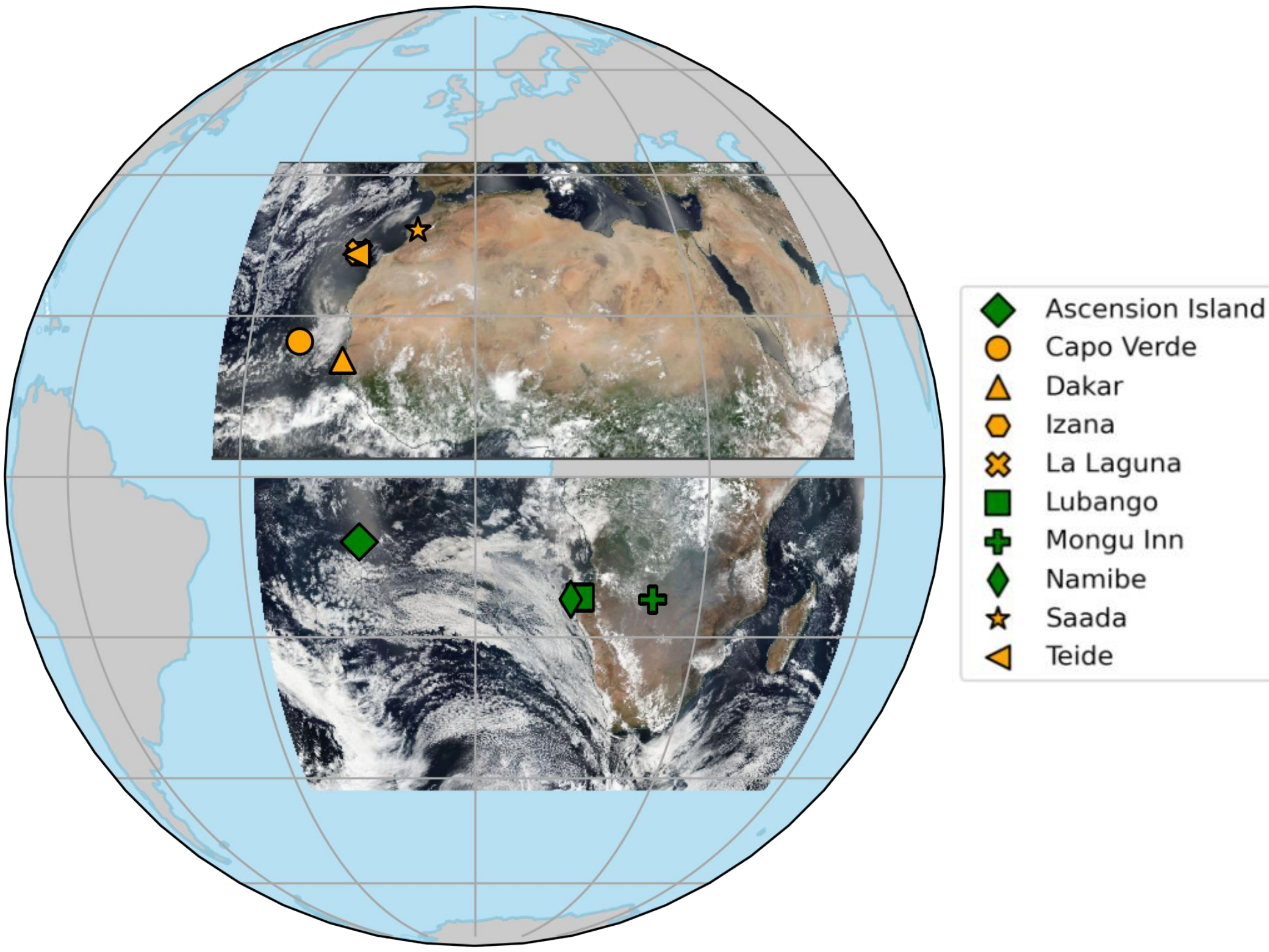

**Figure 3.** Location of AERONET stations considered in this study, along with satellite views of the selected aerosol events taken by the Suomi NPP/VIIRS (Suomi National Polar-orbiting Partnership/Visible Infrared Imaging Radiometer Suite). Stations from CS1 are in orange color and from CS2 in green color. The VIIRS satellite views correspond to (top) 24 June 2016 and (bottom) 26 September 2016 (Credits: NASA Worldview).

### 2.3 Simulation of Synthetic Data

Using synthetic data is essential to evaluate the FCI potential for aerosol remote sensing because it allows us to i) perfectly know the aerosol properties that we aim to retrieve and ii) avoid the impact of unwanted biases (e.g., coming from other variables involved in the inversion). We simulate synthetic FCI-like TOL reflectances for each selected AERONET station and for the duration of the corresponding case study. Satellite reflectance is simulated according to FCI characteristics, by using the corresponding spectral response functions and noise level. In the absence of FCI data for 2016 we use true SEVIRI solar and view angles (identical to FCI's, as MTG-I will eventually be located in the same position of the geostationary Earth orbit than MSG). By doing this, we simulate FCI-like reflectance for the location of each station every 15 minutes. This frequency corresponds to the native temporal resolution from SEVIRI, which is slightly lower than the 10-min temporal resolution from FCI but largely enough to conduct our experiments. Surface reflectance used in simulations is provided from satellite data, with SEVIRI as main data source to make it vary realistically every 15 minutes. Realism is also incorporated into the synthetic data in

terms of aerosol contribution by using 15-min resampled AOD measured by the AERONET stations. Spectral variation of AOD and other optical properties are adopted from realistic aerosol models. To simulate FCI-like data, we use the accurate DOAD solver (De Haan et al., 1987) included in the ARTDECO software (https://www.icare.univ-lille.fr/artdeco/). DOAD solves the radiative transfer equation in the atmosphere by combining successive reflections, back and forth, of the transmission of the radiance through two layers.

More precisely, the FCI simulator developed in this study follows three main steps:

- **Step 1: Scene description:**
  - **Aerosol properties**, which in the case of DOAD correspond to AOD, SSA, extinction coefficient and the scattering matrix. All quantities except for AOD are derived from predefined AERONET-based aerosol models from the MAIAC (Multi-Angle Implementation of Atmospheric Correction)/MODIS C6 aerosol algorithm (Lyapustin et al., 2018). In particular, we choose MAIAC models 6 and 7, giving the microphysical properties of desert dust and biomass burning aerosols, to simulate CS1 and CS2, respectively. All MAIAC models are bimodal, composed of a fine particle mode and a coarse particle mode. The optical properties required by DOAD are calculated for each FCI channel of interest with the MOPSMAP (Modeled optical properties of ensembles of aerosol particles) software (Gasteiger et al., 2018). MOPSMAP also provides for each model the Ångström exponent required to convert the AERONET AOD time series (initially available at 635 nm) to the central wavelength of FCI channels. For simplicity, the resulting aerosol models are named DD (for the coarse-dominated desert dust model) and BB (for the fine-dominated biomass burning model) from this point on.
  - **Surface properties**, which correspond to the bi-directional reflectance of each ground site as seen by Meteosat every 15 minutes. Here, we take into account the diurnal variation of surface reflectance due to the combination of BRDF (Bidirectional Reflectance Distribution Function) effects and the diurnal variation of solar geometry in geostationary observations. Surface reflectance used in this study is derived from satellite data, mainly MSG. More details are available in Appendix A1.
  - **Solar and view geometry**, which is represented by solar zenith angle (SZA), view zenith angle (VZA) and relative azimuth angle (RAA) corresponding to SEVIRI data. Using 15-min SEVIRI angles is essential to add realism to the synthetic data since it helps reproducing the typical geostationary fixed view geometry and diurnally varying sun geometry.
- **Step 2: FCI-like reflectance simulation.** The DOAD solver is chosen due to its accuracy and consideration of light polarization effects, which can be important in the short visible wavelengths corresponding to channel VIS04 for example. While surface anisotropy is taken into account in the simulations by using the varying 15-min bi-directional reflectance of each site, a Lambertian surface is used to account for surface-aerosol multiple scattering. The EUMETSAT official spectral response functions are used to simulate the FCI spectral channels considered in this study, i.e. VIS04, VIS05, VIS06 and NIR22.

- **Step 3: Gaussian noise addition.** This last step aims to make the simulated satellite data more realistic by adding to the synthetic data a Gaussian noise with a standard deviation corresponding to the signal to noise ratio of the FCI spectral channels (i.e., 25 for VIS04, VIS05 and NIR22, and 30 for VIS06; Holmlund et al. (2021)).

The accuracy of the simulator was validated by comparing simulated SEVIRI-like synthetic data with true SEVIRI observations for channel VIS06. Further details on this validation are presented in Appendix A2.

### 2.4 General inversion approach

Inversion of synthetic data is performed using the Levenberg-Marquardt method (Rodgers, 2000), which is accordingly adapted in this study to AOD-only or joint AOD-FMF retrieval. This inversion method is widely used in atmospheric remote sensing, and has been used, for example, to estimate aerosol properties from satellite data including SEVIRI's (Yoshida et al. (2018), Luffarelli et al. (2019), Georgeot et al. (2024)). The Levenberg-Marquardt method is based on the optimal estimation theory (Rodgers, 2000) and its main equation estimates the state vector $x$ at iteration $i+1$ by doing

$$x_{i+1} = x_a + (K_i^T S_\epsilon^{-1} K_i + (1+\gamma) S_a^{-1})^{-1} (K_i^T S_\epsilon^{-1} ((\rho_{\text{obs}} - \rho_{\text{sim}}(x_i)) + K_i(x_i - x_a)) + \gamma S_a^{-1}(x_i - x_a)), \tag{1}$$

where $\rho_{\text{obs}}$ is the satellite observation vector (here in TOL reflectance units), $S_\epsilon$ is the corresponding satellite observation error covariance matrix, $x_a$ is the *a priori* information vector, $S_a$ is the *a priori* covariance matrix, $K_i$ is the Jacobian vector of $x_i$, and $\gamma$ is a damping factor. Quantity $\rho_{\text{sim}}$ is the simulated observation vector, which is in units of reflectance similar to $\rho_{\text{obs}}$. Matrices $S_\epsilon$ and $S_a$ determine how far the estimated satellite reflectance $\rho_{\text{sim}}$ and the estimated solution $x_i$ are allowed to depart from the satellite observation $\rho_{\text{obs}}$ and prior information $x_a$, respectively. Note that vectors and covariance matrices respectively become scalars and variances in the case of estimating AOD only. More details on this inversion method are given in Georgeot et al. (2024).

In this study, $\rho_{\text{sim}}$ is obtained using a fast radiative transfer model (RTM). In most experiments, we use the MSA method (Katsev et al. (2010)) that calculates satellite reflectance based on the well-known Lambertian equivalent reflector approximation (Chandrasekhar (1960)) and analytical equations appropriate for a truncated phase function. The description of this very fast RTM is given in Appendix B along with an assessment of its accuracy in the FCI channels considered in this study using DOAD simulations as reference data. In summary, MSA errors are always below 5% for channels VIS04 and VIS06, whereas they can reach up to 10% for NIR22 in some situations (see Fig. B1).

Another quantity of the optimal estimation theory that is used in this study is the degrees of freedom for signal (DFS, Rodgers (2000)). DFS are widely used to quantify the information content of satellite observations on the variables of interest, and can be calculated according to Coopmann et al. (2020) as

$$\text{DFS} = \frac{K^T * S_a * K}{K^T * S_a * K + S_\epsilon}. \tag{2}$$

It is worth noting that DFS quantify the sensitivity of $\rho_{\mathrm{obs}}$ to $x_i$ based on the corresponding Jacobian vector, but also taking into account the observation and prior covariance matrices.

## 3 Single-channel AOD Retrieval

The first part of the study assesses the potential of FCI visible channels for AOD retrieval. We mainly focus on the comparison of results obtained from the new channel VIS04 with those from the SEVIRI-heritage channel VIS06. Experiments were also conducted for VIS05, but only a brief summary is included for this channel.

### 3.1 Sensitivity Study

We first evaluate the sensitivity to AOD by calculating the corresponding DFS for each observation. We calculate DFS following Eq. 2 for each station included in the FCI-like synthetic data, by setting $S_a$ to 0.05 and $S_\epsilon$ to 0.0001 according to Georgeot et al. (2024).

Figure 4 illustrates the calculation of DFS for sites *Saada* and *Mongu Inn* by showing 15-min time series of AOD, simulated FCI reflectance and DFS, for FCI visible channels VIS04 and VIS06. Results for NIR22 are also shown here, as they will become important in the second part of the study aiming the joint estimation of AOD and FMF (Sect. 4). We can see for *Saada* that AOD is similar for the three spectral channels due to the presence of coarse dust particles. Note the arrival of the dust plume on June 21, with AOD peaking on 25 June with values going beyond 0.6 for all channels. As for *Mongu Inn*, Fig. 4 shows a relatively continuous intense aerosol activity, peaking on September 18 and 28 with AOD reaching a value of 2 in VIS04, with significant spectral variations in AOD due to the presence of fine biomass burning particles.

Figure 4 shows that DFS are the highest in VIS04, identifying this channel as the one with the greatest the sensitivity to AOD. This comes from the lower surface reflectance in the short visible wavelengths (Fig. 2), making aerosol signal predominant in the satellite observations. This is particularly true when AOD is low, as in the first days of the *Saada* time series when minimum values of DFS reach 0.4 for VIS06 and 0.6 for VIS04. However, DFS become similar regardless of channel when AOD is high (e.g., 25 June). Similar results can be observed for the *Mongu Inn* station, which shows DFS values around 0.8 for VIS04 and 0.4 for VIS06. It is important to note the recurrent decrease of DFS around noon in both stations. This variation of DFS during the day comes from the typical diurnal change in AOD sensitivity in geostationary observations due to the changing scattering angle (Ceamanos et al., 2019). For stations located near the prime meridian of the geostationary Earth orbit (e.g., 0° for Meteosat), scattering angle peaks around noon, when aerosol scattering is minimum and surface reflectance is maximum. In our experiments we have observed that this dependency of DFS on scattering angle results in generally higher DFS for CS1 than for CS2 due to the lower scattering angles of the former case study.

As for channel NIR22, we notice a different behavior between the two sites, with a greater difference between VIS04 AOD and NIR22 AOD in *Mongu Inn* due to the presence of fine particles. The low DFS values over this station for NIR22 confirm that AOD sensitivity is almost inexistent for biomass burning aerosols in the near infrared, due to their almost non-existing scattering of radiation at these wavelengths. Some AOD sensitivity however exists in NIR22 in the case of dust over *Saada*,

particularly in the local morning and afternoon when scattering angle is lower. The reason why DFS, and therefore AOD sensitivity, is lower in NIR22 than in VIS04 is that surface reflectance is generally higher in the near infrared (Fig. 2), which is confirmed by the higher TOL reflectance observed for the two stations.

The higher sensitivity to AOD from VIS04 is confirmed in Table 2, which presents mean DFS values for the four selected FCI channels and the two considered case studies (averaged over all stations, see Fig. 3). Results show that DFS is maximum for VIS04 for both case studies, with 0.78 for CS1 and 0.83 for CS2. Contrary to VIS04, DFS are lower in VIS06 and NIR22 for CS2 (with mean values of 0.54 and 0.03, respectively) than for CS1 (0.69 in VIS06 and 0.52 in NIR22). This result is consistent with the strongly decreasing scattering of fine aerosol particles with wavelength (Fig. 2), and can be linked to the limitations for aerosol detection of VIS06 (the main channel used for AOD retrieval from SEVIRI, e.g., Ceamanos et al. (2023)) particularly in the presence of biomass burning aerosols. Results for VIS05 lie in between those obtained for VIS04 and VIS06. Furthermore, Table 2 confirms that channel NIR22 has potential to distinguish between fine and coarse aerosols, as it will be discussed in Sect. 4.

**Table 2.** Mean DFS corresponding to the two case studies, for four different FCI spectral channels.

| Case study | VIS04 | VIS05 | VIS06 | NIR22 |
|---|---|---|---|---|
| 1 | 0.78 | 0.72 | 0.69 | 0.52 |
| 2 | 0.83 | 0.63 | 0.54 | 0.03 |

### 3.2 AOD Inversion

We now investigate if the observed increase in information content with the new FCI visible channels can benefit AOD retrieval, especially in comparison to what is possible with the SEVIRI-heritage channel VIS06. As NIR22 shows no sensitivity to AOD in the presence of fine particles, here we only focus on channels VIS04, VIS05 and VIS06. The inversion approach based on the Levenberg-Marquardt method and the fast MSA (Sect. 2.4) is applied to each channel of the simulated FCI-like synthetic data separately. MOPSMAP-calculated channel-dependent optical properties (Sect. 2.3) for aerosol model DD are used for the processing of the first case study CS1, while aerosol model BB is considered for CS2. The measurement covariance error $S_\epsilon$ is set to 0.0001 according to Georgeot et al. (2024), assuming a similar observation error between SEVIRI and FCI channels. The *a priori* AOD covariance $S_a$ is set to a high value (i.e., 5) because this experiment aims to assess the contribution of each channel information content to estimate AOD, avoiding any interference from prior information. Experiments are conducted in ideal and realistic inversion conditions.

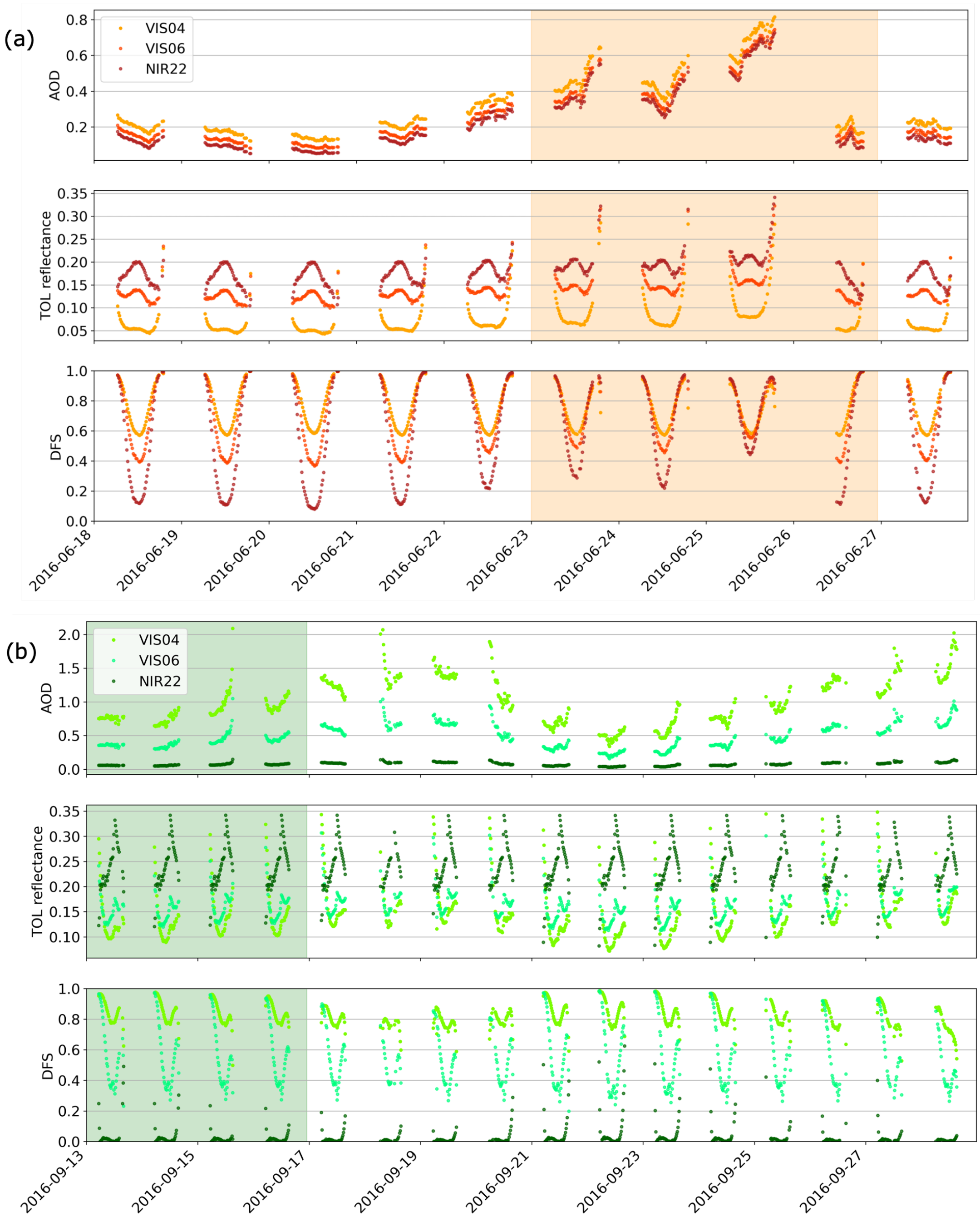

**Figure 4.** Time series of AOD used in the FCI simulator (top panel), simulated FCI-like TOL reflectance (middle panel) and calculated DFS (bottom panel) for FCI channels VIS04, VIS06 and NIR22. These variables are given for stations (a) *Saada* (belonging to CS1) and (b) *Mongu Inn* (belonging to CS2). Shaded days correspond to the dates selected for experiments in Sect. 4.

#### 3.2.1 Ideal Conditions

First, AOD retrieval is carried out in ideal conditions, meaning that all variables required for inversion except for AOD are known. This experiment allows us to quantify the improvements made possible in terms of AOD estimation by FCI new spectral channels without being affected by biases coming from other inputs. It is important to note that inversion results are however subject to several uncertainty sources including the intrinsic biases of MSA, the potential errors coming from the Levenberg-Marquardt inversion, and the Gaussian noise added to the FCI-like observations.

Table 3 compares true and retrieved AOD for the three considered FCI channels. Results confirm the findings of the DFS-based sensitivity study in Sect. 3.1 by showing the highest accuracy for VIS04, with correlation, RMSE (root-mean square error), MBE (mean bias error) and number of retrievals being improved in comparison to VIS06 when the two case studies are considered by +12%, -23%, -65% and +11%, respectively. It is important to remark the significant reduction in MBE, with this score being especially sensitive to the systematic biases affecting MSA (Fig. B1). Again, results obtained for channel VIS05 lie between those obtained for VIS04 and VIS06. Table 3 also stresses the overall better retrieval accuracy for CS1 in comparison to CS2, which can be linked with the more favorable scattering angle in the former case study.

**Table 3.** Mean scores obtained by comparing retrieved AOD to true AOD according to FCI channel: RMSE, MBE, correlation coefficient ($R$), number of retrievals ($N$), mean value of the true AOD ($\tau$) and mean value of the retrieved AOD ($\hat{\tau}$). Scores change with respect to results obtained from channel VIS06 are also shown. Bold numbers are used to highlight the FCI channel with the best retrieval accuracy in each case. Statistics are computed considering the two case studies together (top rows) and separately (middle and bottom rows).

| Channel | Case Study | RMSE | MBE | $R$ | $N$ | $\hat{\tau}$ | $\tau$ |
|---|---|---|---|---|---|---|---|
| VIS06 | 1 and 2 | 0.279 | 0.121 | 0.780 | 3012 | 0.31 | 0.43 |
| VIS05 | 1 and 2 | 0.215 | 0.078 | 0.849 | 2918 | 0.40 | 0.47 |
| VIS04 | 1 and 2 | **0.214** | **0.042** | **0.874** | **3341** | 0.53 | 0.57 |
| w.r.t. VIS06 | | -23% | -65% | +12% | +11% | | |
| VIS06 | 1 | 0.124 | 0.084 | 0.986 | 1905 | 0.27 | 0.35 |
| VIS04 | 1 | **0.044** | **0.022** | **0.990** | **1933** | 0.32 | 0.34 |
| w.r.t. VIS06 | | -65% | -79% | +0.4% | +1.5% | | |
| VIS06 | 2 | 0.431 | 0.185 | 0.510 | 1107 | 0.39 | 0.57 |
| VIS04 | 2 | **0.326** | **0.069** | **0.682** | **1408** | 0.81 | 0.88 |
| w.r.t. VIS06 | | -24% | -63% | +34% | +27% | | |

Figure 5 analyzes further the results by comparing time series of retrieved and true AOD in FCI channels VIS04 and VIS06 for the *Saada* station. This figure confirms that AOD retrieval is more accurate in VIS04, with generally better scores for this

channel. For example, the overestimation of AOD in VIS06 on 18 and 19 June around noon, which is related to the previously discussed limited aerosol sensitivity when AOD is low, is overcome in channel VIS04. Furthermore, the AOD overestimation in VIS06 when AOD is high, from 23 to 25 June, and probably coming from the higher biases of MSA when aerosol load is high (see Fig. B1), is significantly reduced in VIS04 thanks to its generally higher sensitivity to aerosols.

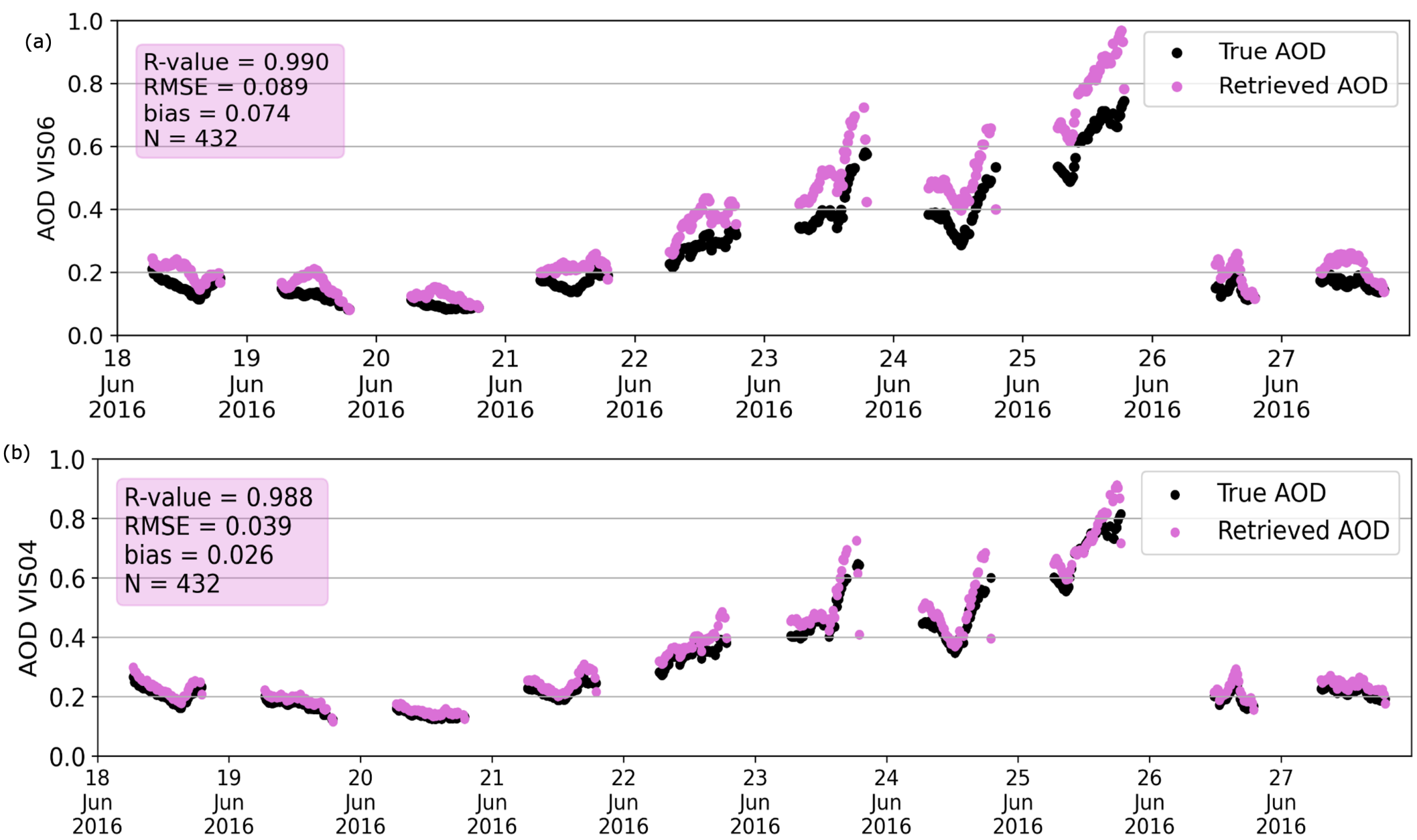

**Figure 5.** Comparison between true AOD (in black color) and retrieved AOD (in pink color) in FCI channels (a) VIS06 and (b) VIS04, for the *Saada* station.

### 3.2.2 Realistic Conditions

We now investigate the FCI sensitivity to AOD in more realistic situations, which means considering some parameters involved in AOD retrieval to be unknown or known with a certain degree of uncertainty. Here, we use the same inversion approach as in Sect. 3.2.1 after adding usual uncertainties to selected parameters. This is done in the following series of experiments:

- **Experiment A:** Similar to the ideal experiment in Sect. 3.2.1, but using a different aerosol model in the retrieval than the one used for simulation. More precisely, the continental Europe (CE) model (corresponding to MAIAC model 4) is used for CS1 instead of the DD model. Analogously, the arid (AR) model (MAIAC model 2) is used for CS2 instead of the BB model. This experiment aims at reproducing the common situation in aerosol remote sensing in which aerosol type is unknown and therefore an incorrect model (and thus inappropriate optical properties) is used for AOD retrieval.
- **Experiment B:** Similar to the ideal experiment in Sect. 3.2.1, but considering optical properties that are slightly different from those of the aerosol model used for simulation. This experiment aims at simulating the case in which an aerosol

model is not totally adapted to the true aerosols over a certain region. For example, smoke aerosol properties change depending on the region due to the presence of different types of combustible (Sayer et al., 2014). To reproduce this situation, we use the usual BB and DD models to invert FCI-like synthetic data simulated with the same aerosol models but with SSA values 5% higher or lower.

- **Experiment C:** Similar to the ideal experiment in Sect. 3.2.1, but adding an error to the surface reflectance used to retrieve AOD. This experiment aims to investigate the common case in which surface brightness is only known with a certain degree of accuracy. In this case, surface reflectance is increased or decreased by 5% when retrieving AOD with respect to the values used in the simulation of synthetic data.

Table 4 summarizes the mean scores obtained for experiments A, B and C. Score change in percentage with respect to the ideal conditions experiment is also shown. Results show that, overall, channel VIS04 is less sensitive to input uncertainties than VIS06. Indeed, scores are generally better when AOD is retrieved in VIS04, including RMSE and MBE despite the usually greater AOD values in this channel. In some situations, the results obtained with VIS04 in the presence of uncertainties are better than those obtained for VIS06 under ideal conditions, which underlines the robustness of this FCI channel. It is worth noting that VIS06's best scores in the case of experiment B with a reduced SSA result from the fortuitous compensation of the MSA systematic biases by the SSA-induced error.

**Table 4.** Scores obtained after assessing the AOD retrieved in the realistic conditions experiments with respect to true AOD, averaged over all stations belonging to the two case studies. Bold numbers are used to highlight the FCI channel with the best results in each experiment.

| | VIS04 | | | | VIS06 | | | |
|---|---|---|---|---|---|---|---|---|
| Experiment | MBE | $R$ | RMSE | $N$ | MBE | R | RMSE | N |
| Ideal conditions | **0.042** | **0.874** | **0.214** | **3341** | 0.121 | 0.780 | 0.279 | 3012 |
| Exp. A: CE | 0.211 | **0.888** | 0.359 | **3335** | **0.148** | 0.725 | **0.352** | 2836 |
| Exp. A: AR | 0.129 | **0.872** | **0.272** | **3370** | **0.123** | 0.715 | 0.285 | 2776 |
| Exp. B: $\omega$ +5% | **0.161** | **0.905** | **0.274** | **3256** | 0.244 | 0.904 | 0.343 | 2804 |
| Exp. B: $\omega$ -5% | -0.068 | **0.827** | 0.234 | **3391** | **-0.025** | 0.662 | **0.205** | 3098 |
| Exp. C: $\rho_s$ -5% | **0.106** | **0.876** | **0.255** | **3366** | 0.223 | 0.826 | 0.324 | 2890 |
| Exp. C: $\rho_s$ +5% | **0.079** | **0.867** | **0.252** | **3266** | 0.091 | 0.711 | 0.285 | 2506 |

Figure 6 further investigates the impact of input uncertainties by showing AOD time series over *Saada* for the ideal conditions experiment and the realistic conditions experiment C, when surface reflectance is decreased of 5%. Results prove that the biased surface brightness has almost no effect on the retrieved AOD in channel VIS04, whereas it results in significant AOD overestimation in VIS06. The biases affecting the red channel come from the lower aerosol information content in VIS06,

particularly around noon. This is less the case of channel VIS04 thanks to its higher AOD sensitivity, which makes it more robust to existing biases such as those affecting surface reflectance or those coming from MSA.

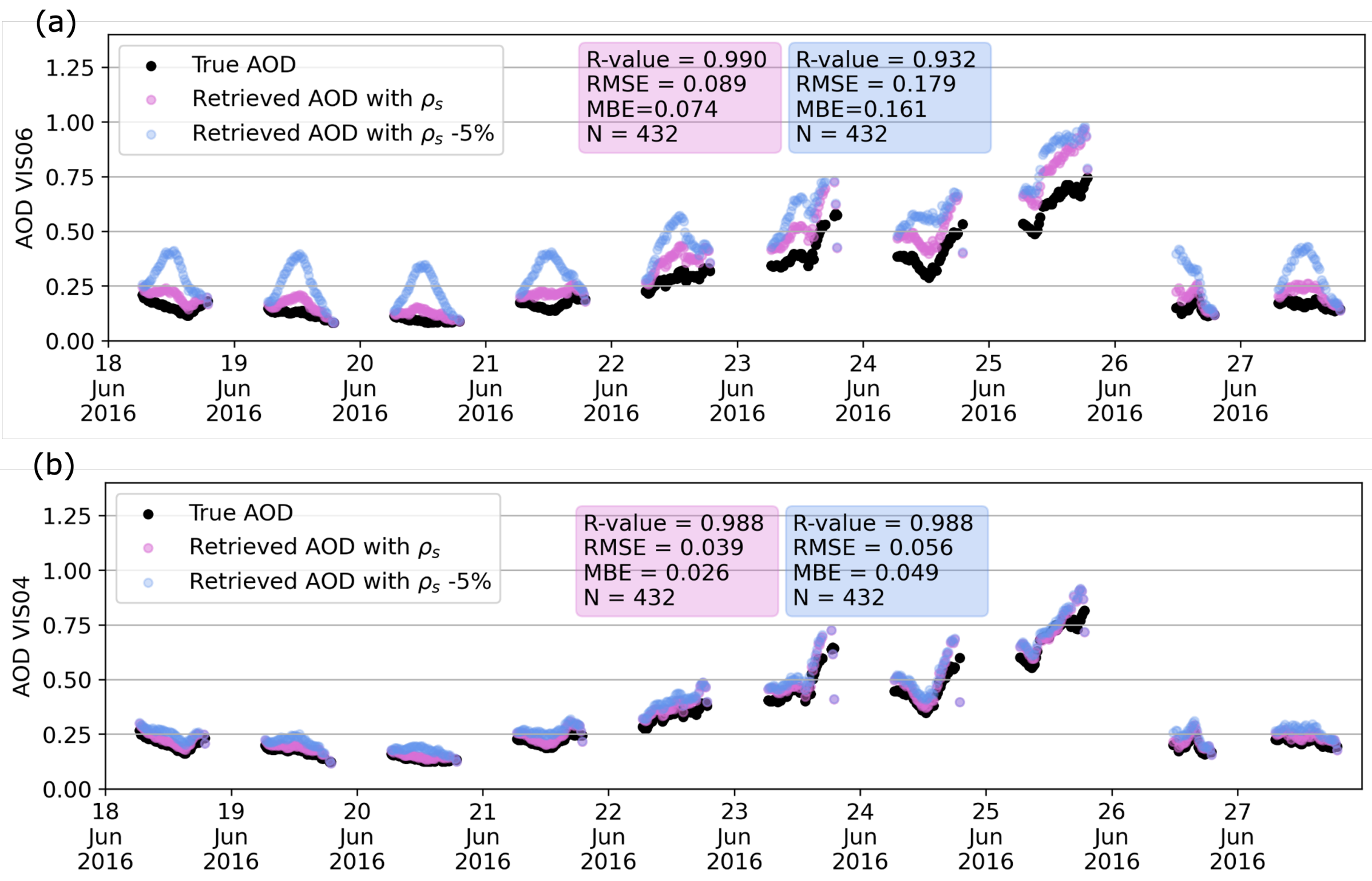

**Figure 6.** True and retrieved AOD obtained after using different surface reflectance in the inversion of the *Saada* station, for FCI channels (a) VIS06 and (b) VIS04.

## 4 Two-channel Joint Retrieval of AOD and FMF

This second part of the study investigates the potential to retrieve both AOD and FMF using the multi-spectral information from FCI. More precisely, we use channel VIS04, which has been proven to be sensitive to AOD in the previous section, and channel NIR22, which contains information on coarse aerosol particles only, as we have seen in Sect. 3.1. This joint retrieval requires to modify the radiative transfer model used for AOD inversion presented in Sect. 2.4, as it is explained in Sect. 4.2. Then, in Sect. 4.3 we do a sensitivity analysis to assess the information content on FMF/AOD in the two selected channels of the FCI-like synthetic data. Finally, we test the joint retrieval of AOD and FMF using two optimal estimation-based methods, a simple one-step approach (Sect. 4.4) and a two-step approach making a smarter use of prior information (Sect. 4.5).

### 4.1 Contribution of Fine and Coarse Modes to AOD

First, we remind that AOD can be separated into the contributions of fine and coarse particles

$$\tau_{\text{aer}}(\lambda) = \tau_{\text{aer,f}}(\lambda) + \tau_{\text{aer,c}}(\lambda), \tag{3}$$

where $\tau_{\text{aer}}$ is the total AOD, $\tau_{\text{aer,f}}$ is the fine mode AOD and $\tau_{\text{aer,c}}$ is the coarse mode AOD.

Fine mode fraction is defined as

$$\text{FMF}(\lambda) = \frac{\tau_{\text{aer,f}}(\lambda)}{\tau_{\text{aer}}(\lambda)}. \tag{4}$$

It is important to note the spectral dependence of AOD and FMF, as aerosol extinction properties depend on wavelength (Fig. 2). For the sake of simplicity, we however omit the dependence on $\lambda$ from this point onward. Furthermore, we refer to the total AOD as $\tau$, and to fine and coarse AOD as $\tau_{\text{f}}$ and $\tau_{\text{c}}$, respectively.

By combining the two equations above, we can express the total AOD as a function of FMF

$$\tau = \tau_{\text{f}} + \tau_{\text{c}} = \text{FMF} * \tau + (1 - \text{FMF}) * \tau. \tag{5}$$

### 4.2 Adapting the RTM to Fine and Coarse Mode Contributions and Hybrid Aerosol model

The MSA radiative transfer model that has been used for AOD inversion simulates satellite reflectance for the selected aerosol model. Although all aerosol models used in this work are bimodal, inversion has been done using the optical properties from the whole aerosol medium, obtained from MOPSMAP, and resulting from the combination of the corresponding fine and coarse modes. Furthermore, the Lambertian equivalent reflector approximation on which MSA is based does not depend on FMF, whereas this is necessary if we want to estimate this variable from satellite data. Hence, the goal here is to express the Lambertian equivalent reflector approximation as a function of (i) the fine and coarse mode contributions to the total satellite reflectance separately and (ii) FMF to be used as weight to combine these two contributions. A similar approach is implemented in Limbacher et al. (2024) to retrieve FMF from geostationary GOES observations.

The Lambertian equivalent reflector approximation (Chandrasekhar (1960)) computes TOL reflectance as the sum of two terms, the additive aerosol contribution and the coupled surface-aerosol contribution

$$\rho_{\text{TOL}} = \rho_{\text{aer}} + \frac{T^{\downarrow}_{\text{aer}} T^{\uparrow}_{\text{aer}}}{1 - a_{\text{aer}} a_{\text{s}}} \rho_{\text{s}}, \tag{6}$$

where $\rho_{\text{aer}}$ is the aerosol reflectance, $T^{\uparrow}_{\text{aer}}$ is the upwelling aerosol transmittance, $T^{\downarrow}_{\text{aer}}$ is the downwelling aerosol transmittance, $a_{\text{aer}}$ is the spherical albedo of the aerosols at illumination from bottom upwards, $a_s$ is the spherical albedo of the surface and $\rho_s$ is the bidirectional surface reflectance. Angular dependency of most terms is omitted here for simplicity.

By separating the single and multiple scattering contributions of aerosol reflectance, Eq. 6 can be expressed as

$$\rho_{\mathrm{TOL}} = \rho_{\mathrm{aer}}^{\mathrm{SS}} + \rho_{\mathrm{aer}}^{\mathrm{MS}} + \rho_{\mathrm{s}}', \tag{7}$$

where $\rho_{\mathrm{aer}}^{\mathrm{SS}}$ and $\rho_{\mathrm{aer}}^{\mathrm{MS}}$ are respectively the single scattering and multiple scattering terms of aerosol reflectance, and $\rho_{\mathrm{s}}'$ is the term depending on surface reflectance.

Next, we express Eq. 7 as a function of the contributions of fine and coarse aerosol modes using the linear mixing method from Wang et al. (1994). This method computes the reflectance of a given aerosol layer as the weighted mean of the individual reflectance values of the coexisting aerosol species, with individual AOD values used as weights. In our case, the linear mixing method is used to consider the coexistence of the fine and coarse modes. Using the linear mixing method, we define the aerosol single scattering term in Eq. 7 as

$$\rho_{\mathrm{aer}}^{\mathrm{SS}} = \frac{\tau_{\mathrm{f}}}{\tau}\rho_{\mathrm{aer,f}}^{\mathrm{SS}}(\tau) + \frac{\tau_{\mathrm{c}}}{\tau}\rho_{\mathrm{aer,c}}^{\mathrm{SS}}(\tau), \tag{8}$$

which becomes after introducing FMF

$$\rho_{\mathrm{aer}}^{\mathrm{SS}} = \mathrm{FMF} * \rho_{\mathrm{aer,f}}^{\mathrm{SS}}(\tau) + (1-\mathrm{FMF}) * \rho_{\mathrm{aer,c}}^{\mathrm{SS}}(\tau). \tag{9}$$

It is important to note that $\rho_{\mathrm{aer,f}}^{\mathrm{SS}}$ and $\rho_{\mathrm{aer,c}}^{\mathrm{SS}}$ depend on the total AOD ($\tau$), and not on the fine or coarse AOD counterparts. Analogously, the surface reflectance dependent term in Eq. 7 is calculated by making

$$\rho_{\mathrm{s}}' = \mathrm{FMF} * \rho_{\mathrm{s,f}}'(\tau) + (1-\mathrm{FMF}) * \rho_{\mathrm{s,c}}'(\tau). \tag{10}$$

Finally, we calculate the multiple scattering term of aerosol reflectance in Eq. 7 according to Abdou et al. (1997), who adapted the linear mixing method to the case of multiple scattering. This is done introducing the fine SSA ($\omega_{\mathrm{f}}$) and coarse SSA ($\omega_{\mathrm{c}}$) to make

$$\rho_{\mathrm{aer}}^{\mathrm{MS}} = \frac{\omega_{\mathrm{mix}}}{\omega_{\mathrm{f}}}\exp^{-\tau_{\mathrm{f}}|\omega_{\mathrm{f}}-\omega_{\mathrm{mix}}|} * \mathrm{FMF} * \rho_{\mathrm{aer,f}}^{\mathrm{MS}}(\tau) + \frac{\omega_{\mathrm{mix}}}{\omega_{\mathrm{c}}}\exp^{-\tau_{\mathrm{c}}|\omega_{\mathrm{c}}-\omega_{\mathrm{mix}}|} * (1-\mathrm{FMF}) * \rho_{\mathrm{aer,c}}^{\mathrm{MS}}(\tau), \tag{11}$$

with $\omega_{\mathrm{mix}}$ being the SSA of the whole bimodal aerosol medium, which can be expressed as a function of FMF as

$$\omega_{\mathrm{mix}} = \mathrm{FMF} * \omega_{\mathrm{f}} + (1-\mathrm{FMF}) * \omega_{\mathrm{c}}. \tag{12}$$

Finally, from Eq. 12, FMF can be expressed as a function of the different SSA

$$\mathrm{FMF} = \frac{\omega_{\mathrm{mix}} - \omega_{\mathrm{c}}}{\omega_{\mathrm{f}} - \omega_{\mathrm{c}}}. \tag{13}$$

Fine (f) and coarse (c) terms of quantities $\rho_{\mathrm{aer}}^{\mathrm{SS}}$, $\rho_{\mathrm{aer}}^{\mathrm{MS}}$ and $\rho_s'$ are calculated using the RTM selected for inversion (e.g., MSA, see equations in Appendix B) fed by the optical properties corresponding to the fine and coarse modes separately, but using the total AOD ($\tau$) according to the linear mixing method.

Joint inversion for AOD and FMF is performed using a bimodal aerosol model that is specifically constructed for this second part of our study. This model, which we name hybrid aerosol model, is built by using the fine mode of the BB model and the coarse mode of the DD model. The hybrid model assigns more or less importance to each one of the modes depending on the FMF value used for inversion, which allows us to process FCI-like observations simulated with the fine mode-dominated BB model or the coarse mode-dominated DD model indistinctly. Again, the required fine and coarse optical properties of the hybrid model are calculated with MOPSMAP based on the corresponding MAIAC microphysical properties (Table 1 in Lyapustin et al. (2018)).

### 4.3 Sensitivity Study

We now investigate the possibilities of retrieving FMF/AOD from FCI channels VIS04 and NIR22 carrying out two experiments. The first one investigates in which conditions and to what extent a changing FMF impacts satellite reflectance. The second experiment quantifies the sensitivity to FMF/AOD by calculating the corresponding DFS.

#### 4.3.1 Impact of a varying FMF on satellite reflectance

We first simulate FCI-like observations for multiple values of FMF in various configurations, with the goal of determining those in which the change in satellite reflectance induced by a varying FMF could be enough to be measured. Simulations are made for a whole day corresponding to stations *Saada* (CS1) and *Mongu Inn* (CS2) using realistic aerosol and surface inputs such as in Sect. 2.3, but using the hybrid aerosol model and the linear mixing method-based RTM approach described in Sect. 4.2.

Figure 7 shows the FCI-like reflectance simulations in four different configurations. First, Fig. 7a shows the diurnal variation of TOL reflectance for *Saada* in channel VIS04, when AOD=1 and for different FMF values, from 0 (pure coarse mode) to 1 (pure fine mode). As it can be seen, FMF variations induce visible changes in VIS04 reflectance, but become less obvious in the middle of the day when aerosol information content is lower. Second, Fig. 7b shows the same simulations but for channel NIR22, which results in larger variations of TOL reflectance according to aerosol particle size, thus confirming the sensitivity of NIR22 to FMF. Third, Fig. 7c shows the same NIR22 simulations over *Saada* but reducing AOD to 0.2, resulting in a significant decrease in FMF sensitivity. Finally, Fig. 7d shows VIS04 reflectance with AOD=1 (similar to Fig. 7a) but for *Mongu Inn*, which corresponds to a less favorable geometry for aerosol remote sensing as previously discussed in Sect. 3. In this case, we observe a decreasing FMF sensitivity during the day due to its dependence to satellite/solar geometry. More precisely, *Mongu Inn* corresponds to more strongly back scattering geometries than *Saada*, with RAA reaching 180° after 3

PM local time. As it will be shown next, information content on FMF is strongly driven by RAA, with higher sensitivity when RAA is close to forward scattering geometries (i.e., RAA close to 0° or 360°) where aerosol scattering is maximum.

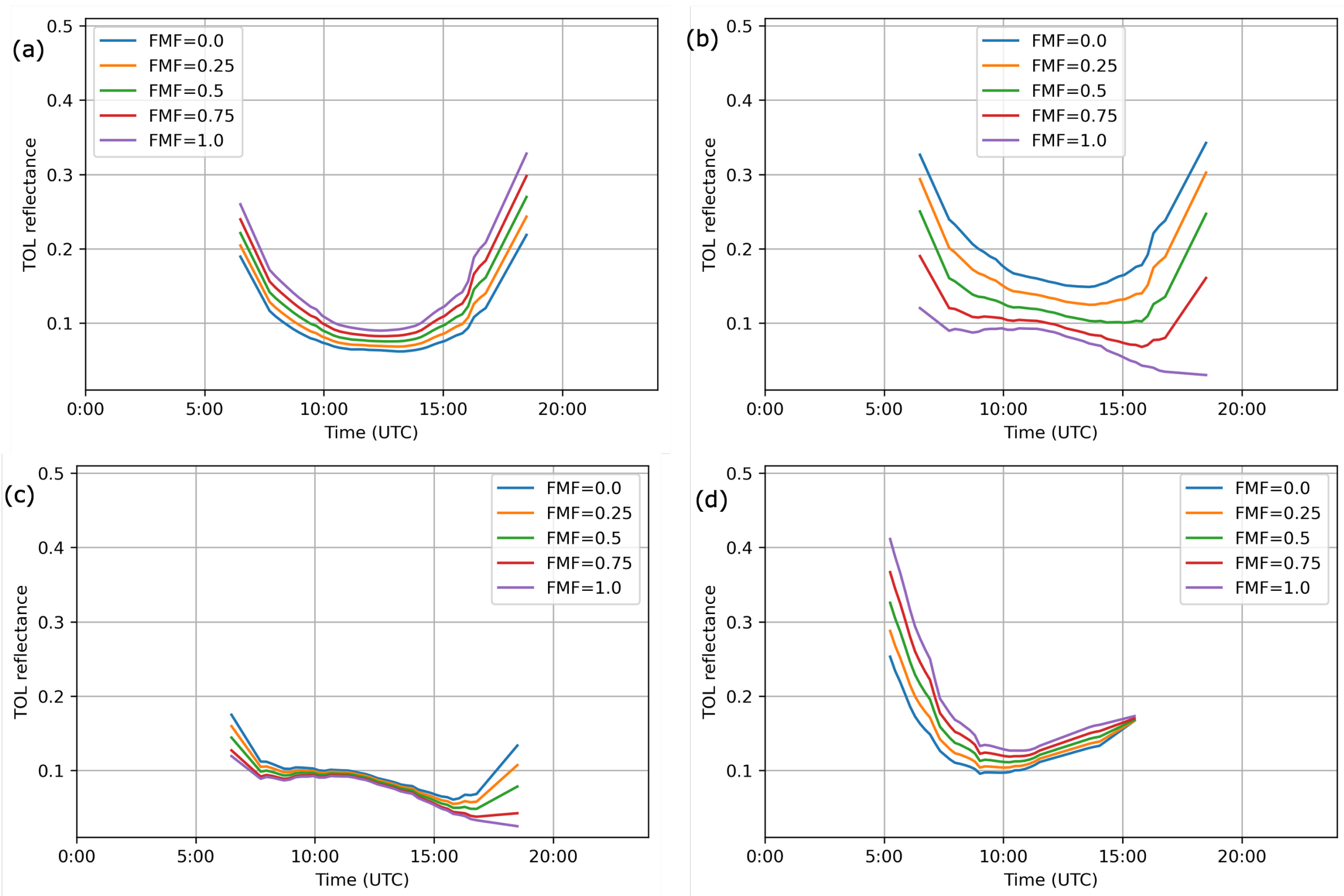

**Figure 7.** FCI-like TOL reflectances for a varying FMF for (a) the *Saada* station in channel VIS04 with AOD=1, (b) the *Saada* station in channel NIR22 with AOD=1, (c) the *Saada* station in channel NIR22 with AOD=0.2 and (d) the *Mongu Inn* station in VIS04 with AOD=1.

### 4.3.2 Degrees of Freedom for Signal in FMF/AOD retrieval

In this second experiment, we calculate DFS in the case of FMF/AOD retrieval from VIS04/NIR22 satellite reflectances. This is done similar to what is done for AOD estimation in Sect. 3.1 but for a 2×2 system (see Eq. 2), making the maximum DFS value equal to 2. Getting close to this value means having a high sensitivity to FMF/AOD, and therefore being likely to estimate the two unknown variables accurately.

Figure 8 shows DFS calculated for varying AOD in *Mongu Inn* (with a fixed VZA of 32°) for a changing solar geometry (with SZA=42° or SZA=70°), a varying surface reflectance (with a vegetation-like dark surface or a barren-like bright surface, see caption of Fig. 8 for more details) and a changing particle size. As for the last parameter, DFS were calculated for fine particles (using BB model, with mean FMF equal to 0.9) or coarse particles (using DD model, with mean FMF equal to 0.2). In this experiment, we also show DFS for different values of RAA to quantify the impact of this angle. Overall, Figure 8 shows

that higher DFS values are reached for low RAA, when aerosol scattering is stronger. Also, the dependency on AOD is obvious, with higher DFS for an increasing AOD until a plateau close to 2 is reached at around AOD=1.5. Regarding the solar geometry, FMF/AOD sensitivity increases with SZA, mostly when RAA is lower than 90°, according to Fig. 8a, with SZA=42°, and Fig. 8b, with SZA=70°. The increase in surface reflectance between Fig. 8a and Fig. 8c results in a decreased FMF/AOD sensitivity especially for low AOD. Finally, we can also notice that DFS are similar when using the BB model (Fig. 8a) or the DD model (Fig. 8d).

In summary, this experiment indicates that the simultaneous estimation of AOD and FMF from FCI observations may become difficult, and therefore prone to estimation errors, in situations with low SZA, low AOD, RAA close to 180° or in the occurrence of bright surfaces. However, other configurations with DFS close to 2 may enable FMF/AOD retrieval, in the presence of predominantly fine or coarse particles.

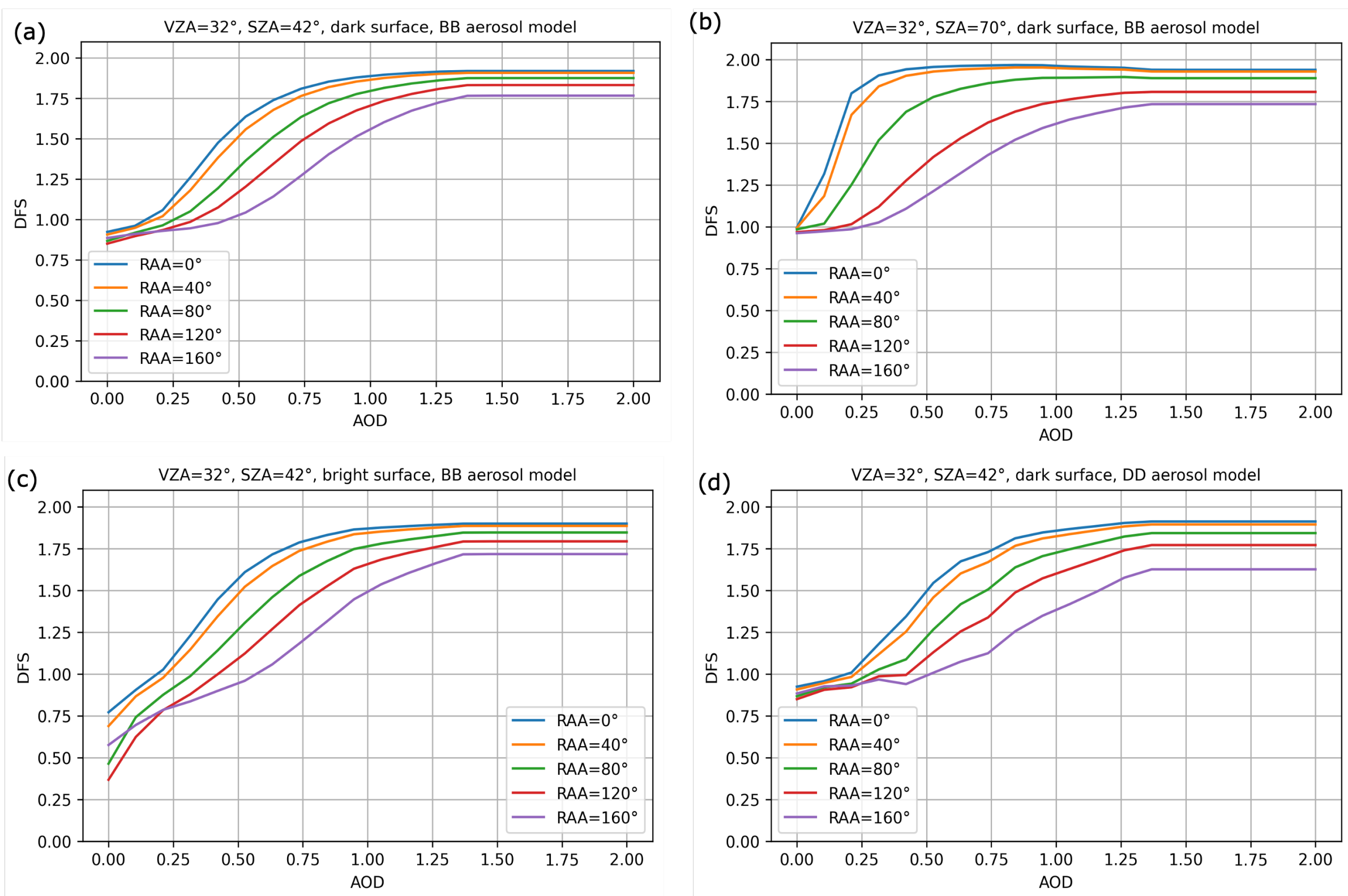

**Figure 8.** DFS calculated according to AOD and RAA for the *Mongu Inn* station (a) for a dark surface with the BB aerosol model and SZA=42°, (b) for a dark surface with the BB model and SZA=70°, (c) for a bright surface with the BB model and SZA=42° and (d) for a dark surface with the DD aerosol model and SZA=42°. Dark surface corresponds to a reflectance of 0.01 in VIS04 and 0.1 in NIR22. Bright surface corresponds to a reflectance of 0.1 in VIS04 and 0.3 in NIR22.

### 4.4 Initial FMF/AOD Retrieval with a One-step Approach

We now assess the findings of the previous sensitivity study by processing FCI-like synthetic data for FMF/AOD retrieval. More precisely, we aim to validate the configurations that were found to be appropriate for this joint retrieval (i.e., those with a DFS value close to 2). This is done by inverting all synthetic data, regardless of the configuration, and checking if the inversion of FMF/AOD is successful or not. Other variables required for the inversion are considered to be known. In order to obtain reliable results, we use the DOAD solver for inversion (i.e., to calculate terms in Eq. 9-11 for each aerosol model) because of its higher accuracy compared to MSA, which shows significant biases in FCI channel NIR22 (see Fig. B1). We pre-calculate DOAD simulations and tabulate them in a look-up-table using appropriate grid sampling following Lyapustin et al. (2011). Linear interpolation according to AOD and surface reflectance is then used for inversion.

In this experiment, we consider time series simulated according to Sect. 2.3 for stations *Saada* and *Mongu Inn* and selected dates (see Fig. 4) only. The true FMF value corresponding to these FCI-like synthetic data is calculated using Eq. 13, which is fed by the SSA values corresponding to the DD model, for *Saada*, and BB model, for *Mongu Inn*, used for simulation.

#### 4.4.1 Inversion Approach

We modify the Levenberg-Marquardt-based inversion approach (see Sect. 2.4) to jointly estimate AOD and FMF in a single step. The goal is to process FCI-like synthetic VIS04 and NIR22 reflectances (i.e., the observations) to retrieved the two variables of interest in channel VIS04. This channel is preferred for the state vector due to its sensitivity to both fine and coarse particles.

The observation vector can be expressed as a function of AOD and FMF at the corresponding channels, making $\rho_{\mathrm{TOL}}^{\mathrm{VIS04}} = f(\tau^{\mathrm{VIS04}}, \mathrm{FMF}^{\mathrm{VIS04}})$ and $\rho_{\mathrm{TOL}}^{\mathrm{NIR22}} = f(\tau^{\mathrm{NIR22}}, \mathrm{FMF}^{\mathrm{NIR22}})$. These two equations are solved using the analytical solution given by Eq. 7 and Eq. 9-11, which uses FMF to weight the contributions from the fine and coarse aerosol modes to the total satellite reflectance (see Sect. 4.2).

The spectral dependence of AOD from the coarse and fine modes are used to express total AOD and FMF in channel NIR22 as a function of their VIS04 counterparts. More precisely, we calculate the fine mode AOD in NIR22 as

$$\tau_f^{\mathrm{NIR22}} = \tau_f^{\mathrm{VIS04}} \frac{2250}{444}^{-A_f^{\mathrm{VIS04,NIR22}}}, \qquad (14)$$

and, analogously, the NIR22 coarse mode AOD as

$$\tau_c^{\mathrm{NIR22}} = \tau_c^{\mathrm{VIS04}} \frac{2250}{444}^{-A_c^{\mathrm{VIS04,NIR22}}}, \qquad (15)$$

where $A_f^{\mathrm{VIS04,NIR22}}$ and $A_c^{\mathrm{VIS04,NIR22}}$ are respectively the MOPSMAP-calculated Ångström exponents, between channels VIS04 (centered at 444 nm) and NIR22 (centered at 2250 nm), for the fine and coarse aerosol modes considered in the inversion (i.e., those from the hybrid model). These two last equations allow us to calculate AOD and FMF for channel NIR22 by simply using Eq. 3 and Eq. 4, respectively.

The resulting analytical expression for VIS04 and NIR22 reflectance enables to estimate the state vector $x_i$ at iteration $i$ (composed of AOD and FMF in channel VIS04) accounting for the fine and coarse modes separately. Again, we use the Levenberg-Marquardt method based on Eq. 1, but using a 2×2 matrix, with $S_a$ and $S_\epsilon$ being covariance matrices this time. Similar to Sect. 3, we set weak *a priori* constraints to be able to explore the true information content of FCI-like data on FMF and AOD.

#### 4.4.2 Results

Figure 9 summarizes the joint FMF/AOD inversion in *Saada* (CS1). Overall, results show a correlation between retrieval accuracy and DFS. Both AOD and FMF in channel VIS04 are indeed estimated quite precisely at the beginning and the end of the day, when DFS are close to 2. During the rest of the day, the lower DFS values correspond to somewhat biased AOD values and clearly incorrect FMF values (equal to 1). The degradation of this 2-variable inversion as we approach local noon is due to the reduced information content coming from an increasingly lower SZA and a RAA more and more close to 180°, which is consistent with our findings in Sect. 4.3. Furthermore, it can be seen how FMF retrieval under favorable geometry is less accurate on 26 June due to a lower AOD value in comparison to previous days. As for CS2, Figure 10 shows similar results for *Mongu Inn* except for the occurrence of high information content (i.e., DFS close to 2) in the morning only due to the different RAA diurnal evolution in comparison to *Saada*. Note the generally lower DFS in *Mongu Inn*, with values going down to 0.8 instead of 1.4 for *Saada*, due to the presence of fine particles invisible in channel NIR22.

It is worth noting that the uncertainties existing in this one-step inversion become much more important in the situations of low sensitivity to FMF/AOD, resulting in the observed incorrect retrievals. Uncertainties include the use for inversion of (i) the Lambertian equivalent reflector approximation combined with the linear mixing method and (ii) the hybrid model, whereas synthetic data were simulated with DOAD and aerosol models BB or DD.

### 4.5 Improving FMF/AOD Retrieval with a Two-step Approach

The previous section proves that FMF/AOD retrieval may be possible from FCI observations in the favorable configurations determined in the sensitivity studies. However, these situations correspond to certain moments of the day only, which does not enable to retrieve FMF during the day at the high temporal resolution made possible by geostationary satellites. We aim to overcome this issue in this section, with the development of a two-step inversion approach exploiting the use of prior information in optimal estimation.

#### 4.5.1 Inversion Approach

The main idea here is to further constrain the data inversion made by the Levenberg-Marquardt method using, as *a priori* information, the AOD and FMF values estimated when FMF/AOD sensitivity is high. The proposed inversion approach, which also uses the hybrid aerosol model, follows two steps:

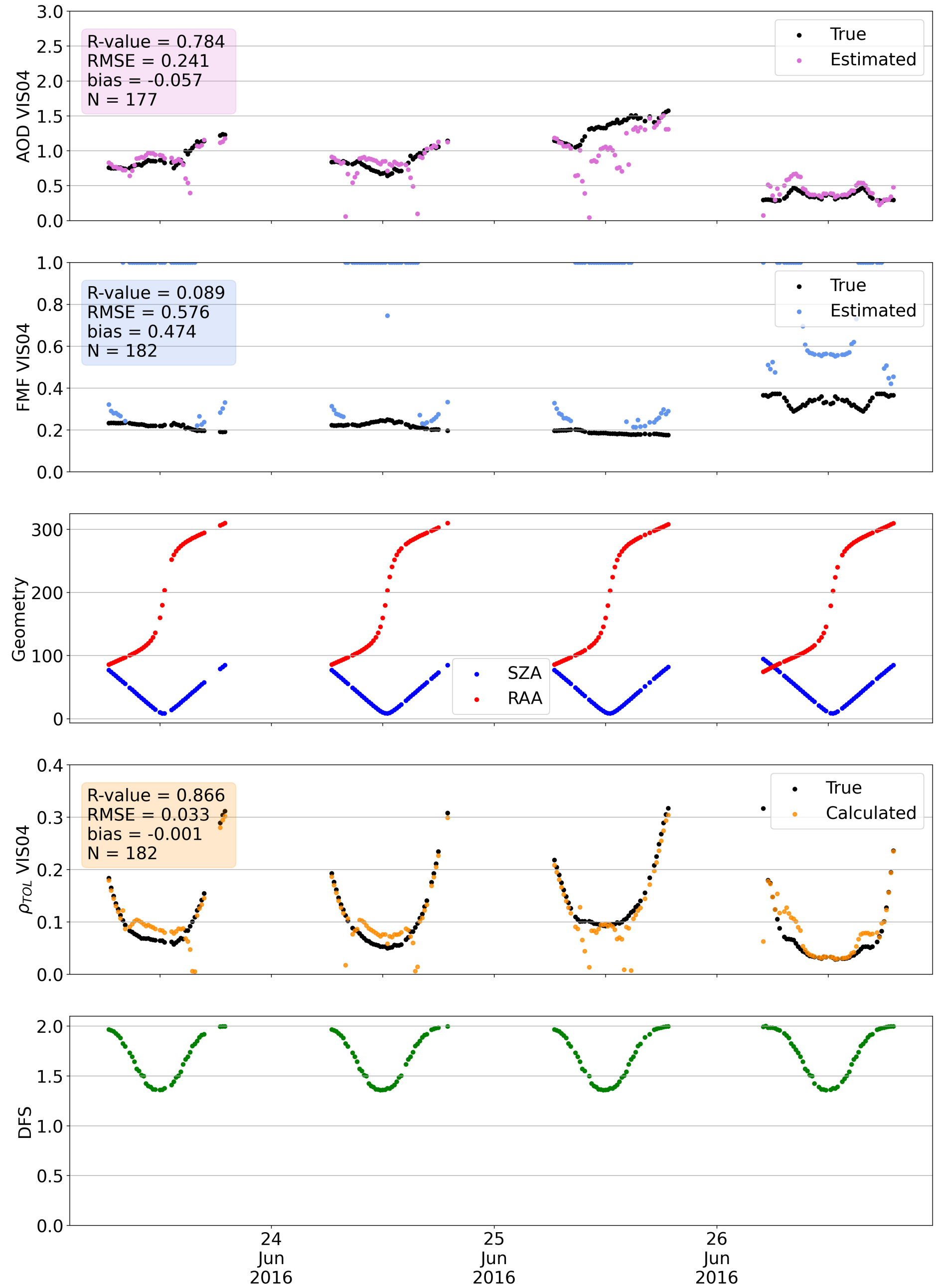

**Figure 9.** Time series of retrieved AOD and FMF, RAA and SZA (in degrees), calculated TOL reflectances and DFS, obtained in channel VIS04 when using DOAD to process the *Saada* station.

– **Step 1:** First, DFS are calculated for all FCI-like observations of a given date. Similar to previous experiments in Sect. 4, the Jacobian vector needed to compute DFS (see Eq. 2) is obtained using prior AOD and FMF values, as true values

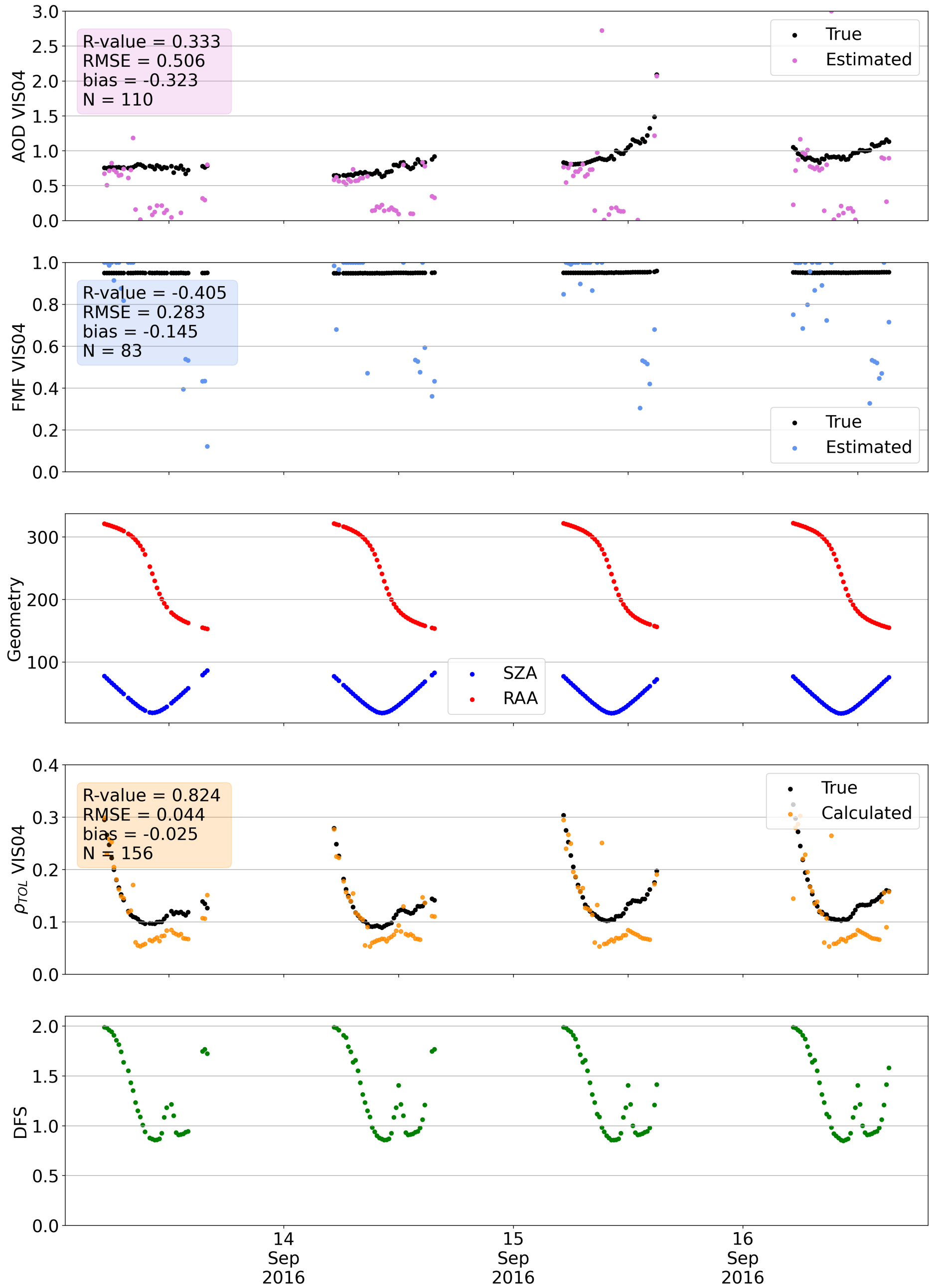

**Figure 10.** Time series of retrieved AOD and FMF, RAA and SZA (in degrees), calculated TOL reflectances and DFS, obtained in channel VIS04 when using DOAD to process the *Mongu Inn* station.

are unknown. In the experiments below, this corresponds to AOD=0.3 and FMF=0.55. These values were found to
495 result in meaningful DFS in most situations. Second, we perform a weakly-constrained (i.e., using a low weight for

prior information) FMF/AOD retrieval from FCI observations with high DFS only (i.e., DFS≥1.95), assuming that information content is enough to jointly retrieve AOD and FMF. Last, we calculate daily averages of the obtained FMF and AOD values.

- **Step 2:** We perform FMF/AOD inversion of all FCI observations using the daily averages from step 1 as *a priori* information ($x_a$). We now give a much higher weight to prior information (e.g., $S_a$ value for FMF 50 times higher than in step 1) such that the Levenberg-Marquardt method uses it as the main data source when FMF/AOD sensitivity is low. By doing this, the inversion method starts the iterative process using more appropriate aerosol properties, which ends up contributing to obtain accurate solutions despite the low information content.

We test this two-step approach using several radiative transfer models (used here to compute terms in Eq. 9-11) with different trade-offs between accuracy and speed in order to assess the possibility of meeting the constraints of operational processing for this joint retrieval. In addition to DOAD (Sect. 2.3) and MSA (Appendix B), we consider another RTM named FLOTSAM (Forward-Lobe Two-Stream Radiance Model; Escribano et al. (2019)). FLOTSAM simulates satellite reflectance based on a two-stream approach and the approximation of the aerosol scattering phase function with a set of basic functions. This allows FLOTSAM to offer a good compromise between speed and accuracy, which makes it appropriate for operational satellite data processing. We determined the approximate computation times per one run of DOAD, MSA and FLOTSAM to be at orders of magnitude of 1 s, $10^{-8}$ s and $10^{-5}$ s, respectively.

#### 4.5.2 Results

Figure 11 shows results obtained with the two-step method using DOAD for *Saada* (simulated with aerosol model DD) and *Mongu Inn* (simulated with aerosol model BB), respectively. First, AOD and FMF in channel VIS04 estimated in the first step (in green and yellow colors, respectively) are generally accurate, except for a few points when AOD is lower (e.g., 26 June for *Saada*), which confirms that they can be trusted as reliable prior information in the second step. Second, final AOD and FMF estimates (in purple and blue colors, respectively) are much smoother and more accurate all day long than with the one-step method, with generally low MBE, low RMSE and high correlation. The slight positive bias (MBE=0.160) affecting the estimated FMF in *Saada* is related to (i) the less good results on June 26 due to the lower AOD and (ii) a general mismatch between the fine mode of the hybrid model used for inversion (that of the BB model) and the fine mode of the DD model used for simulation. This problem does not happen with *Mongu Inn*, with a MBE of -0.019 for the retrieved FMF, due to the strong predominance of fine particles (FMF≈1) that makes the aerosol model used for inversion to be almost identical to the one used for simulation.

It is worth noting that the generally accurate results presented in Fig. 11 have been obtained without any prior information on the aerosol type. Indeed, the use of the hybrid aerosol model allows the Levenberg-Marquardt method to determine the value of FMF (and therefore the particle size distribution) that results in the optical properties leading to an accurate FMF/AOD retrieval.

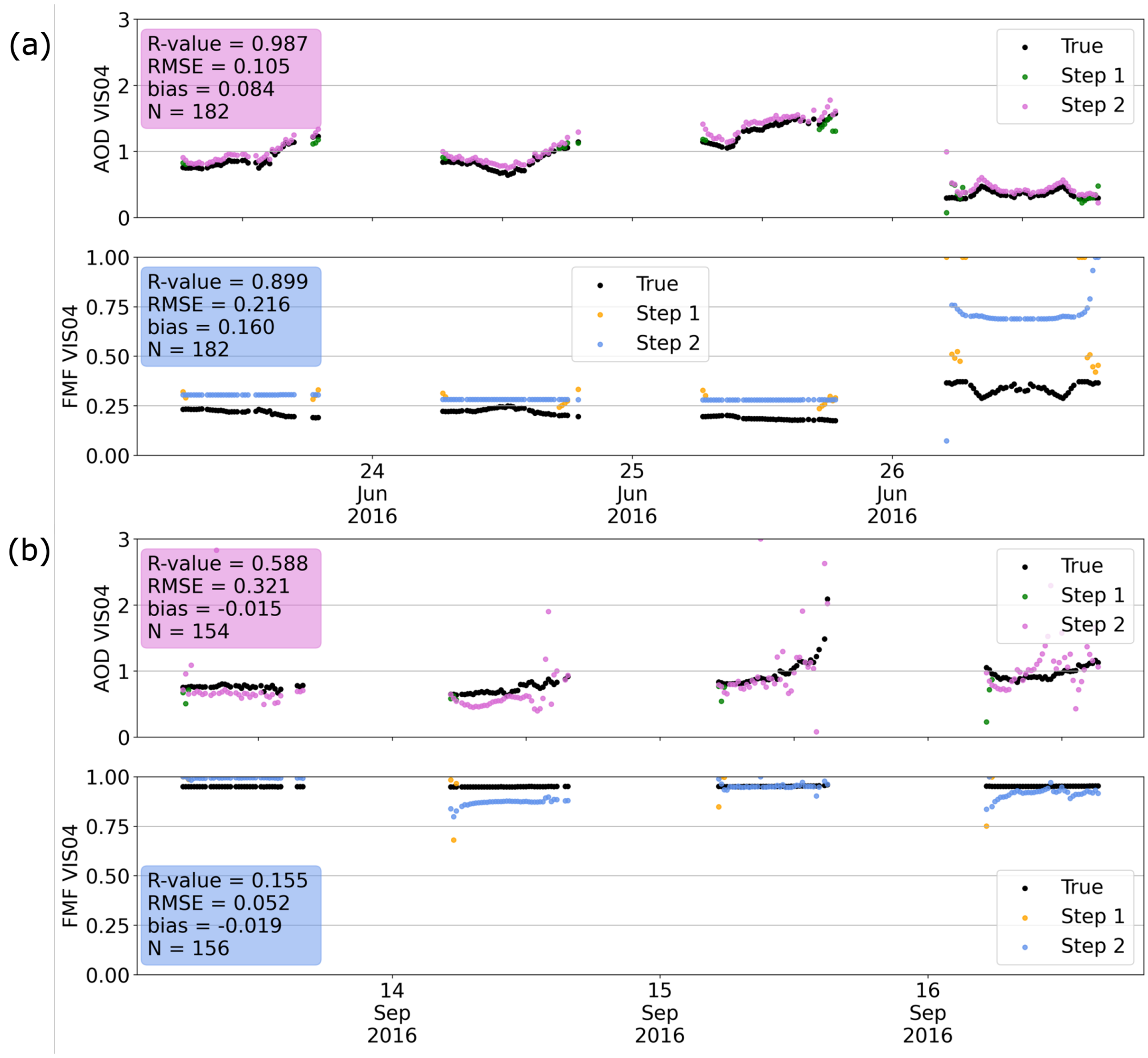

**Figure 11.** Time series of retrieved AOD and FMF in channel VIS04 using the two-step method, obtained when using DOAD for (a) the *Saada* station (b) the *Mongu Inn* station.

Table 5 summarizes the scores obtained for FMF/AOD retrieval in channel VIS04 using the different RTM options (DOAD, MSA and FLOTSAM) in the cases of *Saada* (CS1) and *Mongu Inn* (CS2). The time series corresponding to the MSA and FLOTSAM experiments are shown in Appendix C. Table 5 identifies FLOTSAM as a valid RTM to perform FMF/AOD retrieval. Indeed, FLOTSAM is found to perform generally better than the other fast radiative transfer model MSA (e.g., in terms of retrieved AOD, mainly in *Mongu Inn* with RMSE=0.47 for MSA and RMSE=0.16 for FLOTSAM) and sometimes similarly than the reference model DOAD (e.g., in terms of FMF for both case studies, with slightly higher RMSE for FLOTSAM in comparison to DOAD). MSA is found to provide rather acceptable results for FMF with respect to DOAD, but generally bad results for AOD, which come from the limitations of this fast RTM in FCI channel NIR22 (Fig. B1).

**Table 5.** Average scores of retrieved AOD and FMF in channel VIS04 using three different radiative transfer models and the two-step inversion method, for the *Saada* station simulated with DD aerosols (CS1) and the *Mongu Inn* station simulated with BB aerosols (CS2). Bold text is used to highlight when a given RTM outperforms the rest.

| RTM | Retrieved quantity (in VIS04) | RMSE for CS1/CS2 | Correlation for CS1/CS2 | Bias for CS1/CS2 | N for CS1/CS2 |
|---|---|---|---|---|---|
| DOAD | AOD | **0.10**/0.32 | **0.99**/0.59 | **0.08**/-0.01 | 182/154 |
| | FMF | **0.22**/0.05 | 0.90/0.15 | **0.16**/-0.02 | 182/156 |
| MSA | AOD | 0.24/0.47 | 0.98/0.05 | 0.21/0.12 | 181/143 |
| | FMF | 0.23/0.05 | 0.78/0.60 | 0.18/0.01 | 183/156 |
| FLOTSAM | AOD | 0.21/**0.16** | 0.94/**0.60** | 0.15/-0.03 | 182/156 |
| | FMF | 0.30/0.15 | **0.91/0.62** | 0.21/-0.06 | 182/155 |

## 5 Conclusions

This study showcased the new possibilities in aerosol remote sensing offered by the Flexible Combined Imager on board the EUMETSAT geostationary satellite MTG-I, in particular thanks to its new visible and near infrared channels in comparison to the previous generation of Meteosat satellites with the SEVIRI imager on board. This was achieved in a series of experiments based on realistic FCI-like synthetic data that were generated mimicking true desert dust (case study 1) and biomass burning (case study 2) events in which SEVIRI shows a low sensitivity to aerosols.

In the first part of this study, we assessed the potential of FCI to retrieve AOD and found that channel VIS04 is the most appropriate in both case studies. This was first determined based on the calculation of the degrees of freedom for signal, which were found to decrease with wavelength due to the increasing brightness of land surfaces, combined with the generally decreasing aerosol reflectance (e.g., average DFS equal to 0.83 in VIS04, 0.63 in VIS05 and 0.54 in VIS06 for case study 2). The highest information content of channel VIS04 was then confirmed by a series of experiments in which FCI-like synthetic data were inverted for AOD using an optimal estimation method. The accuracy of the retrieved AOD showed an average improvement of RMSE by -23%, MBE by -65% and correlation by +12% when using channel VIS04 instead of the SEVIRI-heritage channel VIS06. A final experiment considering common uncertainties in the inputs involved in inversion showed that the greater information content of channel VIS04 results in AOD retrievals that are robust against existing biases.

In the second part of the study, we proved that the combination of new FCI channels VIS04 and NIR22 (the latter being sensitive to coarse particles only) enables to go further in aerosol characterization by retrieving fine mode fraction in addition to AOD. First, we investigated the possibilities of FCI-like synthetic observations to achieve this joint retrieval based on the degrees of freedom for signal. Results showed that information content on FMF/AOD is high in situations where aerosol contribution is predominant in the satellite signal (i.e., high AOD, high solar zenith angle, relative azimuth angle far from 180°

and in the occurrence of dark surfaces). This validity domain was confirmed by inverting the synthetic data for FMF/AOD, which resulted in quite accurate retrievals all day along when using a two-step inversion approach that exploits the potential of prior information in optimal estimation methods. It is worth noting that this joint retrieval could benefit from processing additional channels such as done in Limbacher et al. (2024).

In conclusion, this study has demonstrated the promising potential of FCI (operational since December 2024) for aerosol remote sensing. For example, the estimation of FMF (closely related to particle size distribution) opens the door to the determination of aerosol type from Meteosat, which is a step forward compared to the use of climatological information in some SEVIRI-based algorithms (e.g., Ceamanos et al., 2023). Knowing the predominance of fine or coarse particles can indeed be of great interest to monitor the often long ranged transported desert dust and biomass burning smoke plumes (Perry et al. (1997), Ceamanos et al. (2023)). It is worth noting that all results obtained in this work have been validated using fast radiative transfer codes. This makes our findings compatible with operational processing constraints, as FCI's increased temporal and spatial resolutions will require the use of fast methods to process such massive amount of data. Furthermore, our experiments have proved the possibilities of retrieving both AOD and FMF at the sub-hourly high temporal frequency of geostationary imagers such as FCI, which paves the way for the future study of diurnal aerosol variations from space.

## Appendix A: Further Information on the Synthetic Data Generation

### A1 Surface Description

In this study we characterize the surface of the selected scenes simulated in the FCI-like synthetic data by using realistic bidirectional surface reflectance, varying throughout the day, derived from satellite data. The goal here is to have realistic data not only in terms of spectral dependence but also in terms of diurnal variations. This is key to incorporate in the synthetic data the diurnally varying coupling between aerosols and surface brightness at the satellite level, as both contributions generally depend on wavelength and geometry (e.g., through the BRDF in the case of surfaces). The surface reflectance diurnal variations are derived from SEVIRI observations while the spectral dependence, which is necessary to obtain surface reflectance in all the simulated FCI channels, is provided by data derived from the GRASP algorithm applied to the POLDER satellite (Dubovik et al., 2021).

The process is divided into two steps that are applied to each selected station:

– **Step 1: Retrieving diurnally changing surface reflectance.** First, we select a day with a low aerosol load according to AERONET data and we run DOAD to simulate the corresponding 15-min TOL reflectance values corresponding to SEVIRI channel VIS06. This is done using the corresponding 15-min AERONET AOD values as input and considering a totally black surface (i.e., reflectance equal to 0). We then compare the result (corresponding to the aerosol contribution only) with the TOL reflectance obtained from true SEVIRI VIS06 data after correction for gas effects as it is done in Ceamanos et al. (2023). We assume that the difference between these two data sets corresponds to the contribution from the surface to the total satellite signal. With this in mind, we deduce the surface reflectance values (one every 15 minutes)

that cancel out those differences, making the simulations equal to the satellite observations. In a final step, we fill the 15-min gaps due to missing SEVIRI observations with an interpolation method. At the end of this step, we obtain full day 15-min surface reflectance values for the SEVIRI channel VIS06.

– **Step 2: Spectral conversion.** We calculate the 15-min surface reflectance values corresponding to FCI channels VIS04, VIS05 and VIS06 based on spectral ratios derived from monthly GRASP/POLDER surface reflectance data for each station (available at https://www.grasp-open.com/products/polder-data-release/). The POLDER channels used here are centered at 443, 490, 565, 670, 865 and 1020 nm. In order to deal with the differences in central wavelength between SEVIRI, FCI and POLDER, we fit a polynomial curve to GRASP/POLDER data between 300 nm and 900 nm. We then use the fitted curve to calculate spectral ratios between the reflectance values corresponding to the wavelengths of the SEVIRI channel VIS06 and those of the FCI channels of interest. At the end of this step, we obtain 15-min surface reflectance values for the all the selected FCI channels. As for the required surface reflectance for FCI channel NIR22 centered at 2250 nm, we complete our method with GRASP/TROPOMI (TROPOspheric Monitoring Instrument) retrievals, which go further away in the infrared (up to 2300 nm).

It is important to note that this method is based on three assumptions, which we consider to be valid for our simulations. The first one is that surface reflectance follows similar diurnal variation for all channels, because we assume that the shape of BRDF is spectrally invariant (i.e., diurnal variations do not change with wavelength). The second one is that surface BRDF does not change during the selected dates. The third one is that spectral variations are assumed to be the same between POLDER and SEVIRI data despite being acquired in different years (2013 and 2016, respectively).

Figure A1a shows an example of the fitting of GRASP/POLDER data (in blue color) and the surface reflectance deduced from it (in red, green and navy blue colors) for a given 15-min time slot in *Mongu Inn*. Figure A1b shows the diurnal variation of spectral surface reflectance used as input for the FCI-like data simulation in *Saada*.

### A2 Validation of the Simulator

To evaluate the accuracy of the FCI simulator at reproducing geostationary satellite data, SEVIRI simulations are generated and then compared to true SEVIRI observations. FCI data could not be used here because they were unavailable at the time of this study. The SEVIRI channel VIS06, centered at 635 nm, is selected by using the corresponding spectral response function. For this validation, we use the same aerosol, angles and surface input data that we use for FCI simulations. For the sake of consistency, the aerosol properties calculated with the MOPSMAP software are generated at 635 nm.

We address the comparison of SEVIRI VIS06 simulations with true satellite observations for all stations of the two case studies presented in Sect. 2.2. Overall, comparison shows good agreement as it can be seen in Table A1 showing the mean RMSE, correlation, and MBE obtained by comparing the two data sets, for all stations, as well as the number of simulations ($N$). Simulations are found to be accurate, with a mean RMSE of 0.016, a mean MBE of 0.002, and a mean correlation of 0.853. Good results are obtained for all stations, as it is shown by the individual scores, with a slightly higher mean correlation for CS1 in comparison to CS2 (0.892 and 0.800, respectively). However, some differences can be seen for MBE, which is

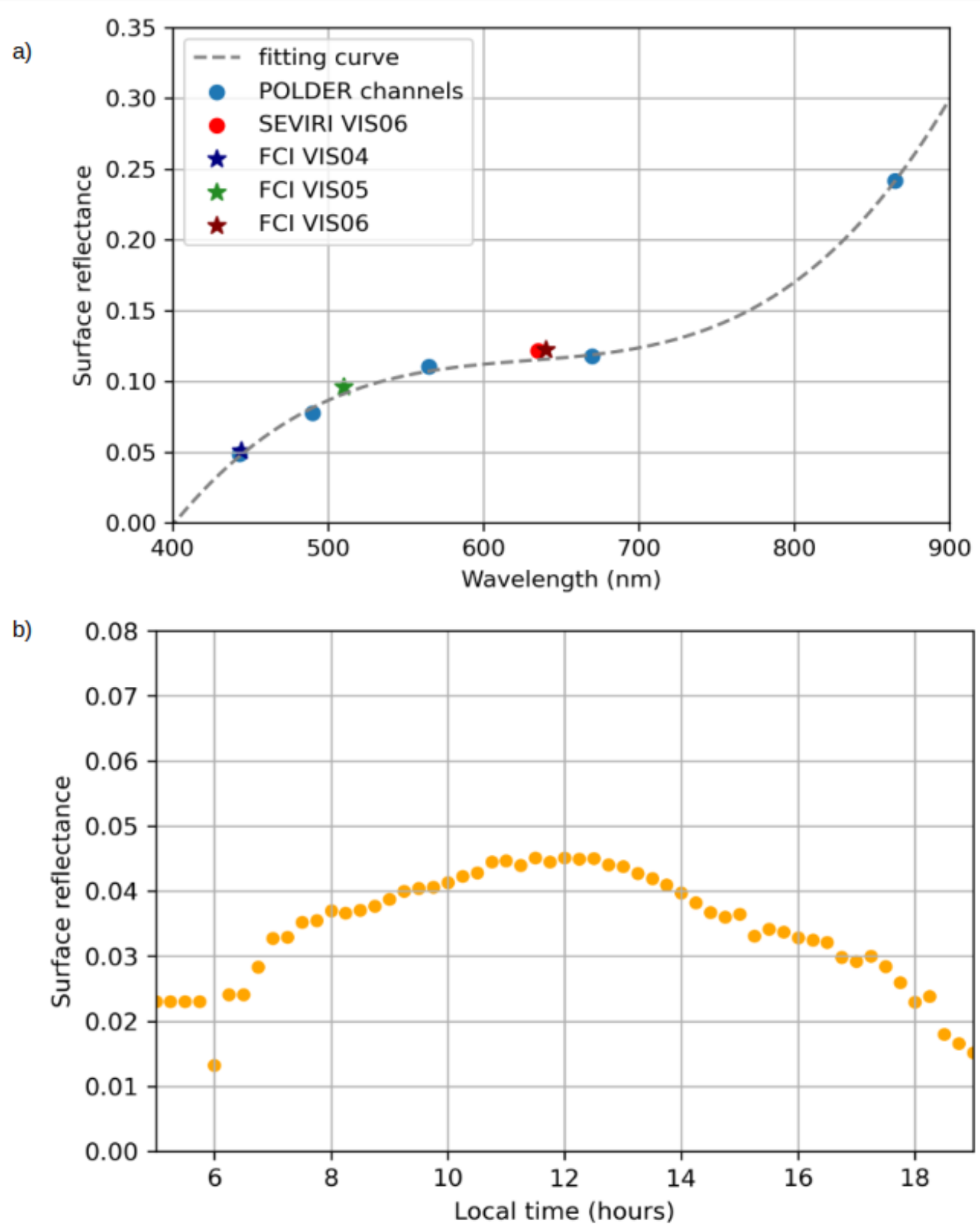

**Figure A1.** (a) Surface reflectance spectral variations for a given SEVIRI geometry at the *Mongu Inn* station in the visible and near-infrared spectrum. The dots correspond to satellite measurements, while the stars correspond to the simulated $\rho_s$ values used by the FCI simulator. (b) Surface reflectance diurnal variations calculated for the *Saada* station in SEVIRI channel VIS06.

always slightly negative for CS2, and slightly positive for all CS1 stations except for *Capo Verde*. This difference can come from the fact that the MAIAC/MODIS aerosol models that are used for the data simulation do not exactly correspond to the true aerosols observed by AERONET.

Figure A2 illustrates the validation of the FCI simulator by showing time series of simulated and observed SEVIRI reflectance for the *Saada* station. In this example, the date of 19 June was selected to estimate surface reflectance. We can notice that the simulator reproduces well the satellite observations, even when AOD is high (particularly on 25 June, with AOD reaching 0.7), and that daily and diurnal variations are well represented. Again, residual errors are assumed to come mainly from residual cloud contamination (e.g., outliers on 21 June) and the fact that aerosol properties used in the simulator (coming from the DD model in the case of *Saada*) do not exactly correspond to the true aerosol properties observed by SEVIRI.

**Table A1.** RMSE, correlation ($R$), MBE and number of simulations ($N$) resulting from comparing SEVIRI observations to SEVIRI simulations, for all the stations selected in the two case studies.

| Station | Case Study | RMSE | $R$ | MBE | $N$ |
|---|---|---|---|---|---|
| *Capo Verde* | 1 | 0.027 | 0.852 | -0.013 | 145 |
| *Dakar* | 1 | 0.013 | 0.909 | 0.003 | 234 |
| *Izana* | 1 | 0.026 | 0.812 | 0.015 | 366 |
| *La Laguna* | 1 | 0.014 | 0.961 | 0.002 | 240 |
| *Saada* | 1 | 0.013 | 0.879 | 0.007 | 408 |
| *Teide* | 1 | 0.018 | 0.940 | 0.009 | 410 |
| **Mean CS1** | **1** | **0.018** | **0.892** | **0.006** | **300** |
| *Ascension Island* | 2 | 0.010 | 0.985 | -0.002 | 31 |
| *Lubango* | 2 | 0.017 | 0.792 | -0.005 | 497 |
| *Mongu Inn* | 2 | 0.015 | 0.788 | -0.003 | 536 |
| *Namibe* | 2 | 0.017 | 0.819 | -0.006 | 251 |
| **Mean CS2** | **2** | **0.016** | **0.800** | **-0.004** | **328** |
| ***Total mean*** | ***1 and 2*** | ***0.016*** | ***0.853*** | ***0.002*** | ***311*** |

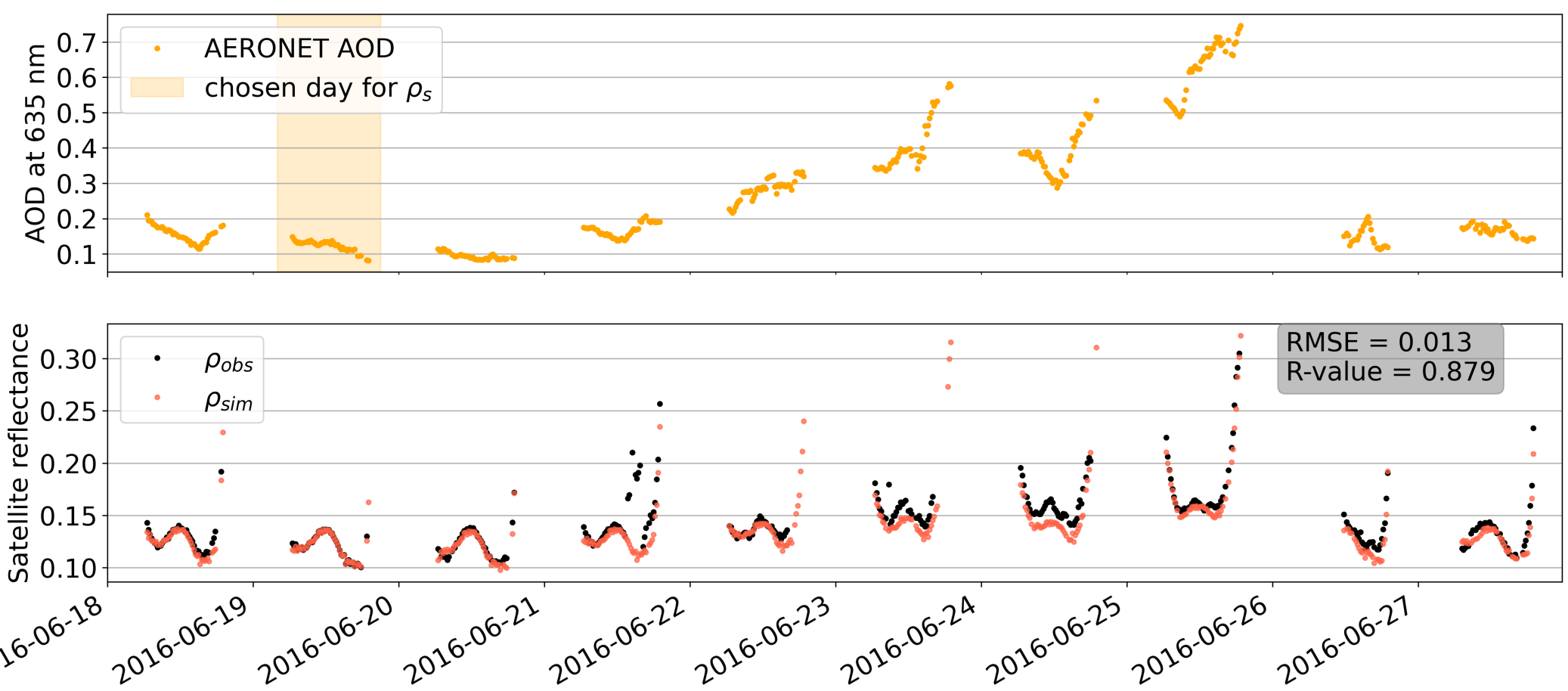

**Figure A2.** Comparison between SEVIRI channel VIS06 observed and simulated TOL reflectance for the *Saada* station (CS1).

## Appendix B: Modified Sobolev Approximation

The Modified Sobolev Approximation (MSA, Katsev et al. (2010)) is a fast radiative transfer model that simulates satellite reflectance combining the Sobolev approximation (Sobolev, 1975) and a truncated phase function, neglecting the lowest scat-

tering angles where aerosol contribution is stronger but satellites never operate at. These features allow MSA to use simple approximated analytical functions that make it very fast. The phase function truncation impose the calculation of aerosol terms in the tilde space, corresponding to the aerosol medium in which the phase function is truncated. In this space, we define $\tilde{\tau}_{\rm aer}$, $\tilde{\omega}$, $\tilde{P}$ and $\tilde{g}$, accounting for tilde AOD, tilde SSA, tilde phase function and tilde asymmetry parameter.

MSA expresses satellite reflectance $\rho$ following the Lambertian equivalent reflector approximation in Eq. 6 and provides the following expressions to calculate all terms in that equation (with $\mu$ terms being the cosine of the corresponding view of solar zenith angles)

$$T^{\uparrow}_{\rm aer}(\mu_v) = \exp[-\frac{\tilde{\tau_{\rm aer}}(1-\tilde{\omega}\tilde{F}_1)}{\mu_v}], \tag{B1}$$

$$T^{\downarrow}_{\rm aer}(\mu_s) = \exp[-\frac{\tilde{\tau}_{\rm aer}(1-\tilde{\omega}\tilde{F}_1)}{\mu_s}], \tag{B2}$$

$$a_{\rm aer} = \frac{\tilde{\tau}_{\rm aer}}{\tilde{\tau}_{aer} + 4/(3-\tilde{x}_1)}, \tag{B3}$$

with $\tilde{F}_1 = 1 - \frac{1-\tilde{g}}{2}$ and $\tilde{x}_1 = 3\tilde{g}$.

Aerosol single scattering is computed by doing

$$\rho^{\rm SS}_{\rm aer}(\mu_v,\mu_v,\phi) = \tilde{\omega}\tilde{P}(\xi)\rho_1, \tag{B4}$$

with $\rho_1 = \frac{1}{4(\mu_v+\mu_s)}(1-e^{-\tilde{\tau}_{\rm aer}m})$ and the air mass being $m = \mu_v^{-1} + \mu_s^{-1}$.

Aerosol multiple scattering is computed by doing

$$\rho^{\rm MS}_{\rm aer}(\mu_s,\mu_v) = 1 - \frac{R(\tilde{\tau}_{\rm aer},\mu_s)R(\tilde{\tau}_{\rm aer},\mu_v)}{4+(3-\tilde{x}_1)\tilde{\tau}_{\rm aer}} + 3[(3+\tilde{x}_1)\mu_s\mu_v - 2(\mu_s+\mu_v)]\rho_1, \tag{B5}$$

with $R(\tilde{\tau},\mu) = 1 + 1.5\mu + (1-1.5\mu)e^{-\tilde{\tau}_{\rm aer}/\mu}$.

MSA is a very fast RTM, with one run being below the range of the microsecond. This allows it to be used in operational inversion algorithms having to process large volumes of satellite data. However, and because of the approximations made to reach this high computational speed, MSA simulations can present significant biases for the retrieval of aerosol properties. Fig. B1 explores the accuracy of MSA for two aerosol models (desert dust and biomass burning) and three FCI spectral channels (VIS04, VIS06 and NIR22). We can see how accuracy varies with AOD, with the highest biases happening for AOD values between 0.25 and 0.75 in most situations. These biases are mostly below 5% for VIS04 and VIS06 but can reach up to 10% for NIR22.

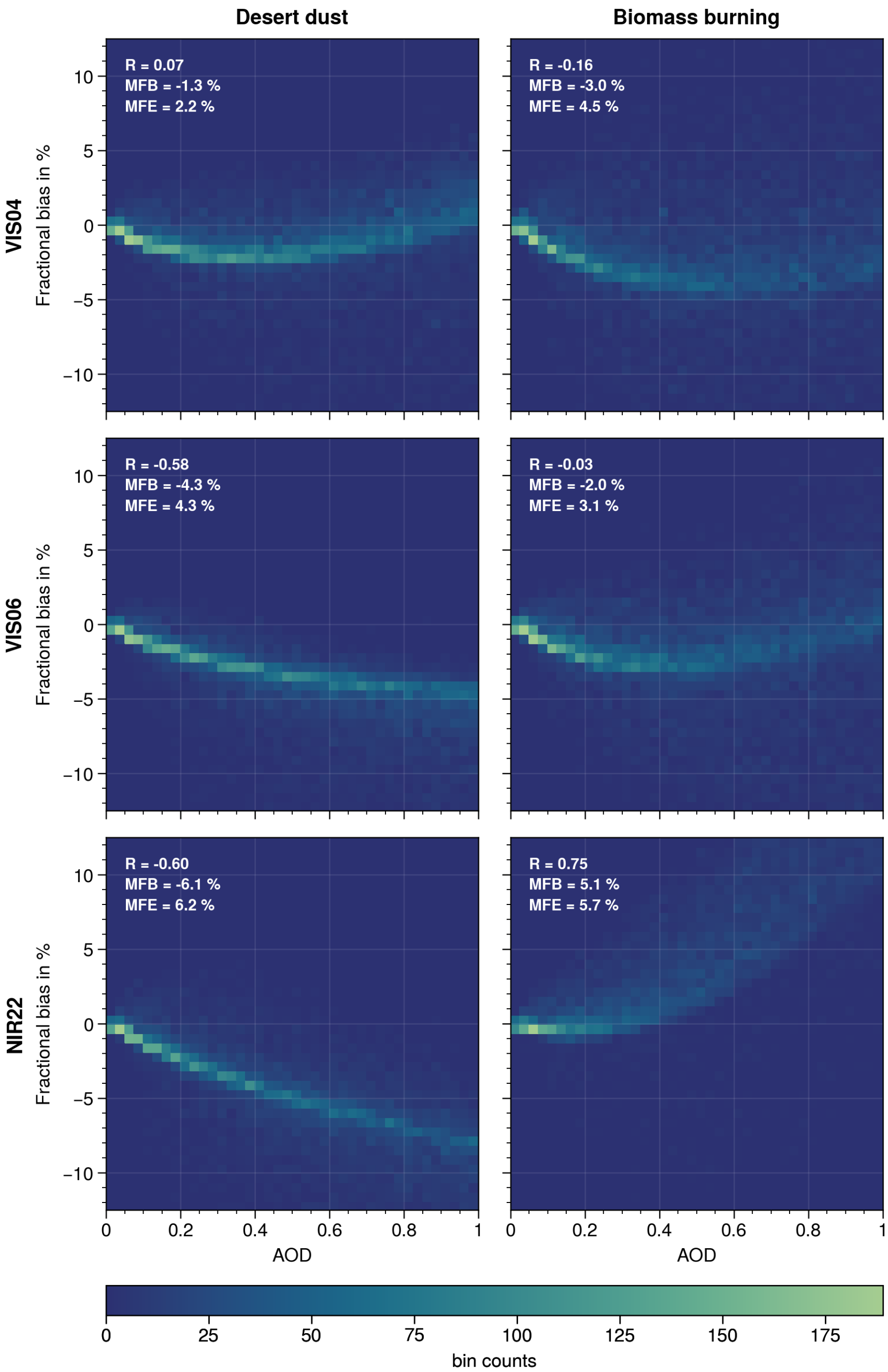

**Figure B1.** Density scatter plots showing MSA accuracy calculated with respect to DOAD, for two aerosol models, desert dust (DD) and biomass burning (BB), according to AOD in FCI channels (a) VIS04 for DD (b) VIS04 for BB (c) VIS06 for DD, (d) VIS06 for BB, (e) NIR22 for DD and (f) NIR22 for BB.

## Appendix C: AOD and FMF Retrieval for Two Additional RTM

We present here the time series of AOD and FMF retrieved when MSA and FLOTSAM radiative transfer codes are used for inversion in Sect. 4.5. Figure C1 shows significant errors in FMF retrieved for *Saada* and AOD retrieved for *Mongu Inn*, which
660 can be attributed to the notable MSA biases in channel NIR22 (see Fig. B1). Figure C2 shows a quite accurate FMF/AOD retrieval when using FLOTSAM, with significant errors only for FMF retrieved on the last day in *Saada* (due to low AOD) and on the second day in *Mongu Inn* (due to an isolated wrong FMF retrieval at the beginning of the day).

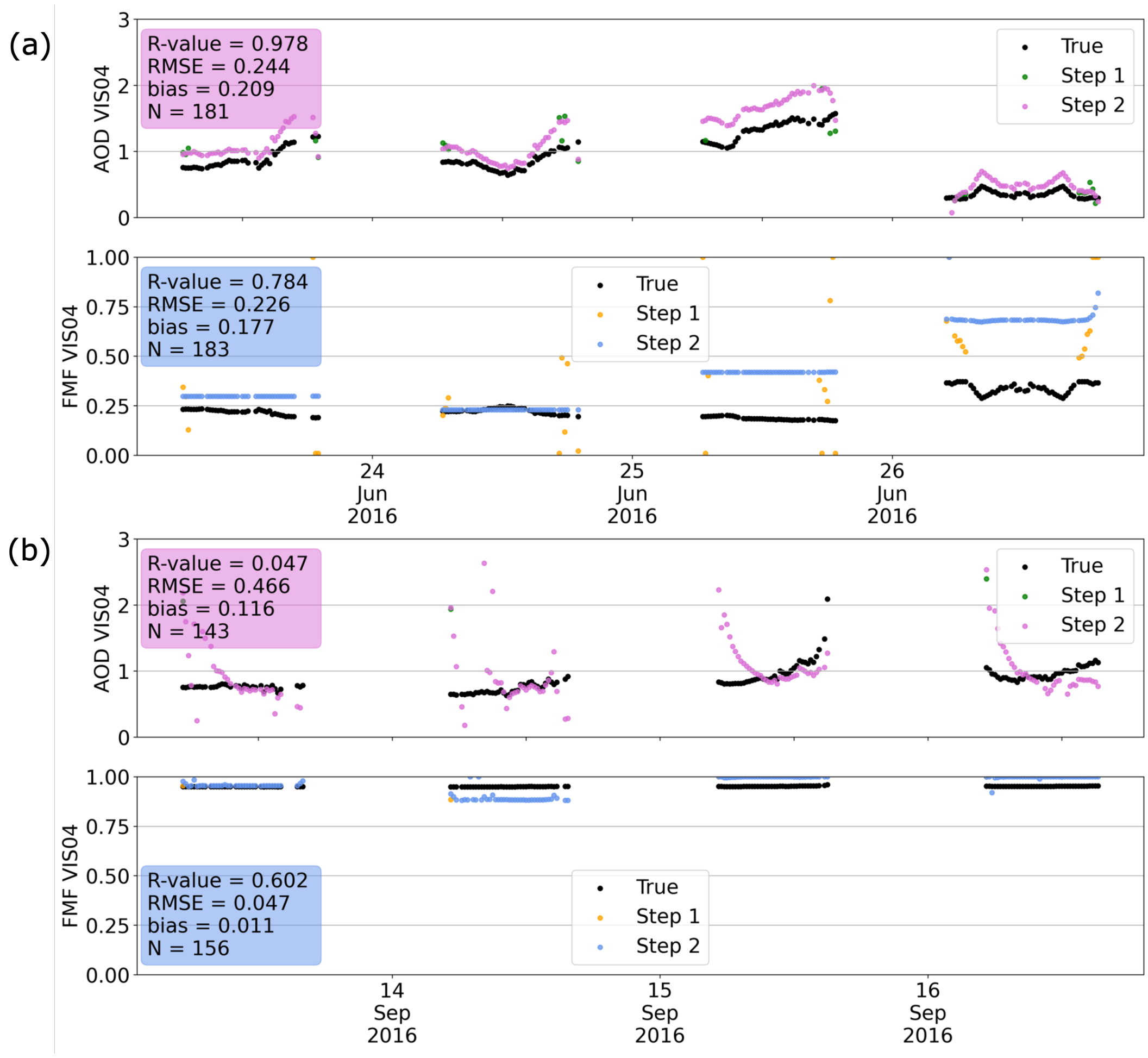

**Figure C1.** Time series of retrieved AOD and FMF in channel VIS04 using the two-step method, obtained when using MSA for (a) the *Saada* station (b) the *Mongu Inn* station.

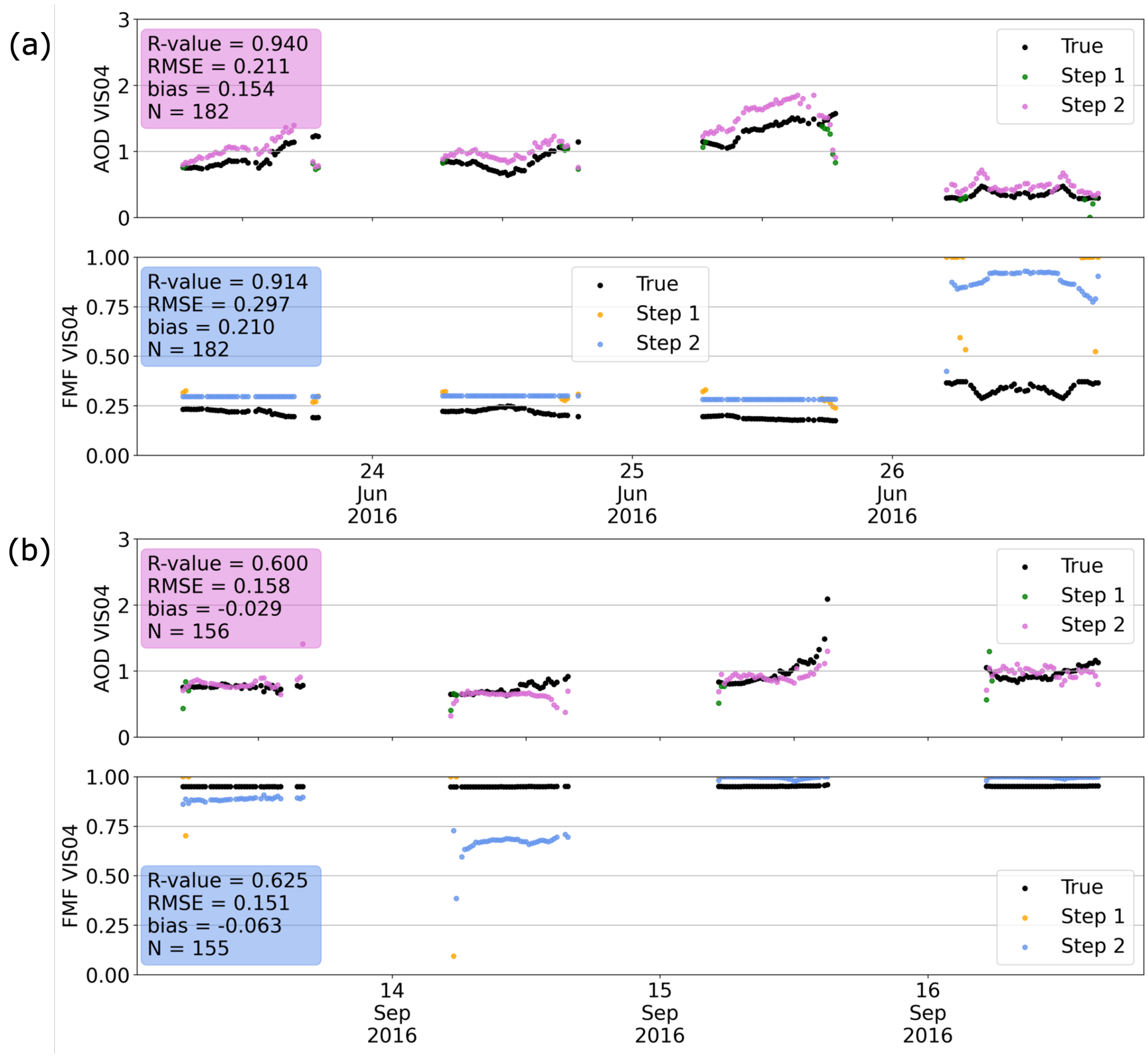

**Figure C2.** Time series of retrieved AOD and FMF in channel VIS04 using the two-step method, obtained when using FLOTSAM for (a) the *Saada* station (b) the *Mongu Inn* station.

*Author contributions.*

AG and XC designed the main ideas of this article and the presented experiments. AG developed the synthetic data simulator
665 with contributions from XC, MC, JG and DJ. AG developed the inversion processes with contributions from XC, JLA and DJ.
AG performed the experiments and wrote the manuscript with contributions from XC, DJ and JLA.

*Competing interests.*

No competing interests are present.

*Acknowledgements.* We acknowledge the use of imagery from the NASA Worldview application in Fig. 3 (https://worldview.earthdata.nasa.gov/), part of the NASA Earth Observing System Data and Information System (EOSDIS). We thank the investigators and their staff for establishing and maintaining the AERONET sites used in this work. ICARE and the SATMOS/CMS center are acknowledged for providing the satellite radiances and geometry.

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
