# Peer review of "Towards improved retrieval of aerosol properties from the geostationary orbit with the new Meteosat Third Generation-Imager satellite"

_EGUsphere, 2025_

## Author Comment (AC1)

**EGUSPHERE-2025-1353**

-

**Response to the reviewers**

Adèle Georgeot and co-authors

June 5, 2025

**Reviewer 1**

*This paper deals with exploring the capabilities in aerosol optical properties retrieval with the Flexible Combined Imager (FCI) onboard the Metosoat Third Generation satellite. The core of the study is to explore the improved retrievals in AOD using the VIS04 channel in FCI and compare the potential of these new retrievals with those in SEVIRI MSG. Authors also explore the capabilities of determining aerosol fine mode fraction using NIR22 channel. The topics are of interest to the scientific community and deserve publication in AMT. However, there are some concerns that need to be addressed before: First, I believe there is essentially nothing new in the retrieval algorithm and the title should be re-though. But most importantly, in my opinion the paper is too lengthy, and it is difficult to catch the main message. I believe that the manuscript can be shortened significantly but keeping the essence of the study. But I understand that the style of the authors must be respected.*

We thank the reviewer for the valuable comments and the time spent reviewing our work. We have attentively addressed all the raised issues and have produced a shorter revised version of the manuscript as suggested. Please find below our replies to the comments and questions from the reviewer (shown in italics). The line numbers given in our answers correspond to the revised manuscript, with the "track changes" mode on.

While it is worth noting that the main purpose of our work is to assess the FCI potential for aerosol retrieval (and not to propose new inversion methods), the reviewer is right when saying that the AOD retrieval algorithm used in our manuscript is not new. Indeed, this corresponds to the method we satisfactorily used in previous works using real GEO data (Ceamanos et al., 2023; Georgeot et al., 2024). However, it must be noted that the AOD/FMF retrieval method contains a certain novelty, mainly in the use of the linear mixing model to account for fine and coarse modes. We agree that the initial title could be a bit misleading, as both AOD and FMF are variables that are commonly retrieved from satellite missions, mainly LEO-based. This is however not the case for geostationary missions, and Meteosat in particular, which makes AOD/FMF retrieval novel in this context. We have therefore modified the title accordingly as it follows: "Towards improved retrieval of aerosol properties **from the geostationary orbit** with the new Meteosat Third Generation-Imager geostationary satellite".

Regarding the length of the manuscript, we have produced a shorter version as suggested by the reviewer by removing some parts of the initial manuscript (e.g., Figure 1, Sect. 4.3.1, equations in Appendix B) and moving some others to the appendices (e.g., Sect. 2.3, Sect. 4.1 and Sect 4.2). With this, and after revising many parts of the manuscript to improve its clarity, we believe that the revised manuscript enables to catch the main messages more easily now, while keeping the essential information.

*Apart of my previous concerns related to the shape and structure of the manuscript, I believe that there is not enough discussion of the sensitivity of the retrieval to errors in the input optical data – or at least it as not been clear for me -. For example, I see many diurnal patterns in TOL, AOD... that seems to follow the pattern in SZA and I wonder if there is an artifact in the retrievals that amplifies errors. I also believe that the assumptions in surface and aerosol models are too strong when going to the real world. This should be at least discussed, although it is true that the information content for*

*aerosols is low in the FCI when compared with more advanced imagers/polarimeters.*

We acknowledge the existence of retrieval errors that depend on SZA (and therefore the time of the day in GEO observations) in some of our experiments based on synthetic data. According to our experience, this is something expected in AOD retrieval from real GEO satellite data due to the variation of the AOD information content during the day. This diurnal variation actually comes from the diurnal change in solar geometry, which results in a diurnally varying aerosol scattering (peaking when SZA is high, and therefore scattering angle is low; eg. at the beginning and end of the day over regions along the 0° meridian) and an also diurnally varying surface reflectance (peaking when SZA is low, and therefore scattering angle is high; eg. around local noon along the 0° meridian, see TOL reflectance curves in Fig. 3a). As a result, AOD retrieval becomes more difficult at noon over Western Europe, for example, due to the combination of a weak aerosol signal and a much stronger surface reflectance, which in turn amplifies any bias existing in the input data. This explanation was added in L321 of the revised manuscript. It must be noted that some of our experiments, other parameters such as surface reflectance do not introduce errors in the retrieval process because they are known in the retrieval (e.g. Sect. 3.2.1; in this case, the retrieval error comes from other sources, and can be amplified by the diurnally changing AOD information content). In experiments in Sect. 3.2.2, however, some parameters are intentionally biased before their use in the retrieval to mimic more realistic situations in satellite remote sensing, thus impacting the quality of AOD retrieval.

As for the assumptions on aerosol model and surface reflectance, experiments in Sect. 3.3.2 aim to get closer to real world conditions by assessing the impact of using some "wrong" parameters (ie. different from the input data used for simulation) for inversion. In particular, experiments A and B consider wrong aerosol properties, while experiment C considers a wrong surface reflectance. In this section, we observe less good results than in the "ideal" case (Sect. 3.2.1) mainly for FCI channel VIS06, while results are still good for channel VIS04 thanks to its increased information content. In our opinion, these experiments show that it should still be possible to obtain accurate information on aerosols with real FCI data even if surface and aerosol models are not perfectly known. According to the reviewer suggestion, we have added a new paragraph on this point in the conclusions: see L493 "It must be noted that the retrieval accuracies observed in this study may be degraded when processing real FCI data, mainly due to differences between the optical properties of the aerosol models used for inversion and those of the real world aerosols observed by FCI (e.g., showing a large variability of mixtures in the case of desert dust and biomass burning particles). In the near future, the inversion methods used in this work will be tested with actual FCI data and real world limitations will be determined. For example, the consistency of the hybrid model used for FMF/AOD inversion with respect to the natural variability of smoke and dust particles will be assessed. Furthermore, the use of daily averages of retrieved AOD and FMF to constraint the second inversion of the two-step method will have to be circumvented if we are to meet near real time constraints."

**MINOR REVISIONS**

- *Line 36: It is not true that you try to address particle size distribution. You are trying to estimate fine mode fraction. Please correct.* This was corrected.

- *Line 48: FCI is not constantly covering the entire Earth, just part of it. Please correct.* This was corrected.

- *Figure 1: It is not discussed in the text. What is the point of including Figure 1.* Figure 1 was removed.

- *Figure 2: Biomass-burning and dessert dust are illustrated as aerosol types. But in the real world, there are large variability and mixtures of biomass-burning and dessert dust. Assuming all aerosols within these types have the same properties is too simple, although it helps in satellite retrievals. I recommend being careful in the discussions.* We agree that our dust and smoke models are a simplification of real world conditions, which we made in our study for the sake of simplicity. The larger variability of real aerosols and the potential impact of not accounting for it is now mentioned in a new paragraph in the conclusions (see L493). It is worth noting, however, that this point was addressed when adding uncertainties in the retrieval process in Section 3.2.2. For example, we have seen in Exp. B that modifying the single scattering albedo (and therefore

considering that the observed smoke or dust do not correspond to the aerosol model used for inversion) does affect the quality of AOD retrieval, but much less in channel VIS04 than in channel VIS06. Similar results are obtained in Exp. A when a completely different aerosol model is used.

- *Line 139: I do not understand the expression 'satellite synoptic'.* The term "synoptic" was removed.

- *Section 2.3 Simulation of Synthetic Data: I strongly recommend using a box-chart to help the reader better understand the different steps.* A new figure has been added (see Fig. A1).

- *Line 147: What is TOL? Please define the acronyms at least the first time they appear. Similar happen to others (i.e. MSA, DFS, SZA, RAA). Please, be careful.* We have added a table of acronyms in Appendix E and made sure all of them are explained when they first appear in the text.

- *Section 3.1 Sensitivity Study: To me, a sensitivity study is to study the impact of random and systematic errors in the retrievals. Maybe it is necessary to rephrase this section.* This word usage was incorrect and has been corrected. Now, all occurrences of "sensitivity study" in the manuscript have been renamed "information Content Analysis".

- *Line 259: I do not see clearly how Table 2 confirms NIR22 potential to distinguish between fine and coarse mode. Can you specify ?* According to the reviewer comments, this was clarified by adding the following sentences L251: "as in this channel DFS is much higher for coarse aerosols (predominant in CS1) than for fine aerosols (predominant in CS2), respectively, 0.52 versus 0.03. The latter value confirms the almost inexistent information content on fine particles in NIR22, which means that an AOD signal retrieved in this channel will very likely mean that the observed aerosols consist of coarse particles"

- *Table 3: What is w.r.t VIS06?* This was corrected, and replaced by "diff. to VIS06", meaning the scores difference with respect to VIS06. This is now explained in the caption of the table.

- *Table 5: Can you explain better the first column? What is w?* This was corrected, and $\omega$ was replaced by the acronym of single scattering albedo (i.e. SSA).

- *Section 4.5.1: I wonder how you are going to apply your inversion approach to real FCI data. Can you explain something?* We have added some discussion about this subject in the "Conclusions" section (L495): "In the near future, the inversion methods used in this work will be tested with actual FCI data and real world limitations will be determined. For example, the consistency of the hybrid model used for inversion with respect to the natural variability of smoke and dust particles will be assessed. Furthermore, the use of daily averages of retrieved AOD and FMF to constraint the second inversion of the two-step method will have to be circumvented if we are to meet near real time constraints."

---

## Author Comment (AC2)

**EGUSPHERE-2025-1353**

-

**Response to the reviewers**

Adèle Georgeot and co-authors

June 5, 2025

**Reviewer 2**

*This manuscript explores the additional information content gained from the new channels for the Flexible Combined Imager (FCI) on the Meteosat Third Generation-Imager compared to its predecessor, the Spinning Enhanced Visible and Infrared Imager (SEVIRI) on the Meteosat Second Generation platform. It uses synthetic top of layer reflectances generated using a combination of aerosol models using known exact atmospheric and solar and sensor geometric parameters as well as surface reflectance data derived from monthly GRASP/POLDER surface reflectance data which have been curve fit to the satellite spectral bands and then calculated at 15-minute intervals for two challenging case study regions, one in a desert dust event and another in a biomass burning event. An information content analysis is performed on retrievals for both AOD-only (to compare FCI with SERVIR) and AOD+Fine Mode Fraction (to highlight the potential for FCI to further characterize aerosol properties with the new channels).*

We thank the reviewer for the valuable comments and the time spent reviewing our work. We have attentively considered all the raised issues and have produced a revised version of the manuscript addressing most of them. The manuscript has been restructured and significantly revised to improve its clarity and completeness, and it has also been shortened (as suggested by other reviewers) but keeping the essential information. Please find below our replies to the comments and questions from the reviewer (shown in italics). The line numbers given in our answers correspond to the revised manuscript, with the "track changes" mode on.

**General Comments:**

*The topic itself is incredibly relevant and the scientific community can stand to gain a lot from it, but I feel that this manuscript is not finished in terms of characterizing the uncertainties and utilizing them in their sensitivity analyses for retrievals using the different channels, particularly over the course of the diel cycle and at different AOD concentrations. The assumptions of constant error covariance matrices for both the input satellite observations as well as the a priori inputs to the aerosol models seems overly simplistic, and I worry about their impacts on the final results, especially as these are very complex aerosol case studies. I'd like to see if both the final retrievals and the sensitivity analysis significantly changes due to a dynamically-varying uncertainty at the very least for:*

- *the trace gas correction as a function of solar and sensor angles,*

- *the polynomial fit function to interpolate the GRASP/POLDER spectra to the FCI channels,*

- *a systemic bias due to the magnitude of the AOD retrieval itself as an a priori,*

- *the instrument uncertainty itself as a function of wavelength ( 5% for the VIS and 10% for the NIR), and*

- *a spectrally-invariant BRDF shape as a function of solar angle*

While we agree that the definition of appropriate covariance matrices is important for accurate inversion in optimal estimation, we honestly think that the additional work suggested by the reviewer is out of the scope of the present manuscript. Our study focuses on the potential of the new FCI for aerosol remote sensing, which is addressed by comprehensive analyses of (i) the information content of new measuring channels and (ii) their performances for AOD and FMF retrieval in comparison to SEVIRI's. To achieve this goal, we have used inversion methods that were satisfactorily used in previous works to process real SEVIRI data for AOD retrieval (Ceamanos et al., 2023; Georgeot et al. 2024). The choice of using existing methods (including their setup in terms of covariance matrices) was made to focus the present manuscript on the FCI potential assessment, since the development (and validation!) of new inversion methods is a topic in itself that deserves fully-dedicated works in our opinion. Despite assuming constant error covariance matrices in AOD retrieval, the selected inversion methods were comprehensively validated in the previously cited past works by successful comparisons of the SEVIRI-obtained AOD retrievals with reference data, including AERONET measurements and state-of-the-art satellite products. For example, Ceamanos et al, 2023 analyzed around 400,000 AOD retrievals over the whole SEVIRI field of view to find a very good agreement with AERONET (i.e., MBE=0.010, RMSE=0.093, R=0.800), comparable to that found for the state-of-the-art product GRASP/POLDER (see Fig. 13 in Ceamanos et al., 2023). For us, this is proof enough that the selected inversion methods, which make assumptions of constant covariance matrices, can allow us to achieve the main goal of the present manuscript (i.e. the assessment of FCI potential for aerosol remote sensing). This is now clearly mentioned in the new text starting at L214. While we agree that dynamically-varying uncertainties could benefit aerosol inversion in some situations, the accurate calibration of the covariance matrices according to all parameters suggested by the reviewer would require extensive experiments that do not fit in the present manuscript and therefore should be addressed in a separate study.

It is also worth noting that gas correction was not carried out in our study, as FCI data were simulated in top-of-layer (TOL) reflectance units (i.e., including aerosol and surface contribution, but excluding gas effects) for the sake of simplicity (see L127: "Synthetic data are simulated without including atmospheric gases for the sake of simplicity and to focus on the retrieval of aerosol properties. This choice results in the simulation of top of aerosol layer (TOL) reflectance, resulting from the contributions from aerosols and the surface only. "). Correction of FCI observations for gas contribution is being currently addressed in our team and will be published in a separate publication.

As for the surface reflectance used in the synthetic data simulation, the main goal was to consider the natural variation of surface reflectance in GEO observations coming from the combination of the surface BRDF and the changing solar geometry during the day (every 15 min in our study). This was made possible by estimating for each selected station 15-min surface reflectance from real SEVIRI VIS06 observations, which was then satisfactorily validated as presented in the appendices (e.g., see Table A1). It must be noted that GRASP/POLDER data were only used in a second time to derive the spectral ratios used to "scale" the SEVIRI-derived 15-min surface reflectance spectrally (as indicated now in L594), from the SEVIRI VIS06 channel to each of the considered FCI channels. The goal here was to obtain a realistic spectral variation of surface reflectance (which is proved to be essential in aerosol satellite retrieval in our work). Therefore, the potential uncertainties coming from the GRASP/POLDER-based polynomial fit are not relevant as long as the resulting spectral surface reflectance captures well the natural spectral variation, which is proved to be the case. In fact, residual uncertainties are not important in the data inversion, as the final surface reflectance (including those uncertainties) is considered to be the "true surface" in our synthetic observations. The same goes for the spectrally-invariant BRDF shape, which we assume to be a valid assumption to simulate our synthetic data. Similar to the GRASP/POLDER spectral scaling, this assumption with respect to "the real world" does not affect the inversion results in our study as the resulting surface reflectance becomes the true surface in "the synthetic data world". We added a line on our confidence on the surface reflectance used in the data simulation in L601: "Despite these assumptions, the realism of surface reflectance obtained with the presented approach is judged to be high overall".

Finally, we did not consider the currently known channel-dependent FCI uncertainties (as this reviewer mentioned, 5% for the VIS and 10% for the NIR) because these values were not known when the study was carried out. Instead, we used the expected SNR values (25 for channels VIS04, VIS05 and NIR22 and 30 for channel VIS06 according to Holmlund et al. 2021) to define the Gaussian noise that was added to the synthetic data. This spectral change in SNR was not accounted in the definition of the inversion uncertainties for simplicity, but also because it was considered to be weak (n.b., SNR

only differs for channel VIS06). As explained in L214 (and L262, L343), the observation error is set for all channels to 0.0001, which roughly corresponds to an uncertainty of 3% (and therefore a SNR of 30). A new line was included in the revised manuscript on this point (L214: "For example, the value of $S_\epsilon$ was set to encompass the uncertainty of the SEVIRI channel VIS06 (equal to 3% according to Luffarelli et al. (2019)) and other errors operating in the inversion process.")

*This might be outside the scope of this manuscript as it stands, but while I do really like the evaluation of the information content changes for extreme aerosol events like desert dust and biomass burning, I was left wanting to see it also in the context of more "typical" retrievals like standard atmospheres over land and/or ocean to give a larger perspective on how much more information content we can gain from FCI in a more global context.*

We agree that knowing the FCI performances for "typical" aerosol conditions is also important, but we preferred to conduct our study on more extreme and therefore more retrieval-challenging aerosol conditions for which the previous SEVIRI instrument showed limitations. While Ceamanos et al. (2023) and Georgeot et al. (2024) showed a good agreement of SEVIRI AOD retrievals with reference data in regions with standard aerosol conditions (e.g., Europe, or South America when fire smoke is not present), they found much greater errors in bright surface regions such as Northwest Africa (corresponding to case study 1 in the present work) and in some areas of Southwest Africa due to aerosol-unfavorable geostationary satellite geometry (corresponding to case study 2). Naturally, AOD retrieval over more "aerosol-standard" regions (i.e., with an already acceptable AOD information content from SEVIRI) will improve with FCI thanks to the general increase in AOD sensitivity made possible by this new sensor.

The following sentence on this subject has been included in the "Conclusions" section (L480): "It is worth noting that the FCI performances for AOD retrieval in less extreme and therefore less retrieval-challenging aerosol conditions than the ones considered in the selected case studies (e.g., anthropogenically influenced continental aerosols) are also expected to improve with respect to SEVIRI's."

**Specific Comments:**

- *Line 84 (and throughout the manuscript): I think it might be better to call the channels by their center wavelengths (e.g. "FCI 444 nm channel") throughout the text, or at the very least have a table with the center bands and the FWHM (but I'd much prefer having it in the text to avoid the reader having to keep referencing it).* We have added a new table in the introduction specifying the names of the spectral channels of FCI and SEVIRI, as well as the characteristics relevant in our study (see Table 1).

- *Figure 1: This figure by itself seems like it could be removed for brevity, especially as the FCI data were still pre-operational. More attention then could be devoted to Figure 2 to highlight the FCI additions.* The figure has been removed as suggested.

- *Figure 2: Could we change the SEVIRI bands to grey (or something slightly different from the gridlines) and the FCI to blue to highlight them as the newer wavelengths? Also, the "Sand" color and the "Desert Dust" color are blending to my eyes (especially with the right legend overlaid so closely to the desert dust line). Could this be swapped to another color, and the spectral reflectance lines given a different weight or dotted/dash style to discriminate them better?* The figure was modified according to the reviewer's suggestion, and vertical grid lines were removed for more clarity. *This figure could also be partnered with a table going over the wavelength comparisons between FCI and SEVIRI (showing channel names, band centers, FWHM, etc).* We thank the reviewer for this idea. As suggested, we have added a new table in the introduction (see Table 1).

- *Section 3.1: This section might be better titled "Information Content Analysis" since it primarily analyzes the DFS which isn't quite the same as the uncertainties of the retrievals.* We have renamed the section "Information Content Analysis".

- *Line 225: I wonder if this is an oversimplification to be set as constants when it could vary spatially and temporally depending on the input models (such as the polynomial fit uncertainty*

*for the spectra, or the uncertainty from trace gases especially at high angles, the biases in retrievals due to AOD magnitude, etc). How would the impacts from these affect the information content retrievals performed here?* Please refer to our previous first answer to this point above.

- *Table 2: Instead of simply calling it case study 1 and 2, for clarity it might be better to call it dust and biomass burning.* After some thinking, we have preferred to keep calling them "case study 1" and "case study 2". In our opinion, using "desert dust" and "biomass burning" would create confusion with the aerosol models that we used in our work, which are named the same.

- *Figure 4: All three orange-red markers as well as the light green markers are very difficult to distinguish here.* We have modified the markers colors in the two sub-figures to make them more visible. We also checked that markers are easily distinguished in the black-and-white version of the figure.

- *Line 338: Would a systemic bias in TOL reflectance from this influence the final results?* Here we meant that we choose to omit the dependence of equations on lambda for the sake of simplicity (i.e., to simplify the writing of the equations to come). However, the spectral dependence of all variables was taken into account in all the experiments of our study.

- *Line 464: What were the values for these? Are they based on the Georgeot et al 2024 paper?* Values are Sa=(0.2, 0 & 0, 0.5) and Sy=(0.0001, 0.00052884 & 0.00052884, 0.0001). The variance for TOL reflectance in channels VIS04 and NIR22 was chosen accordingly to Georgeot et al., 2024. The covariance values for TOL reflectance in VIS04 and NIR22 were calculated by using the synthetic data set and computing the covariance between these two channels. We have added explanations in the revised manuscript on this point (L343 "The measurement covariance matrix $S_\epsilon$ is set as follows: $S_\epsilon = \begin{pmatrix} 0.0001 & 0.00052884 \\ 0.00052884 & 0.0001 \end{pmatrix}$. The variance (0.0001) for channels VIS04 and NIR22 was chosen accordingly to Sect. 3. The covariance values (0.00052884) are obtained from the whole synthetic data set by calculating the covariance between the TOL reflectance of these two channels. As for the a priori covariance matrix, it is set to $\begin{pmatrix} 0.2 & 0.0 \\ 0.0 & 0.5 \end{pmatrix}$ following a similar empirical approach as in Georgeot et al. (2024)."

- *Line 500: A 50x multiplier seems very arbitrary. What would different Sa values do for final retrievals?* This value was selected empirically after carrying out several experiments testing different values of Sa, which sets the weight given to the prior value used for FMF. Lower (higher) Sa values were found to constrain too much (not enough) the retrieved FMF, and resulted in less accurate retrieved AOD time series for example.

- *Line 585: Could you add a bit more detail about this correction in the text here? Just enough to cover how it's done for the full diurnal timeframe.* The following sentence was included in L578 "More precisely, satellite radiance was corrected for gas absorption and Rayleigh scattering considering a US Standard 62 atmosphere adjusted by model analyses of surface pressure, total column water vapor and total column ozone.". *As they use SMAC to correct for molecular effects, what are the minimum and maximum ranges of solar and sensor angles for these stations? How are the increased uncertainties here at sunrise and sunset accounted for in the sensitivity analysis?* Details on the gas correction made with SMAC can be found in Ceamanos et al. 2021 and 2023, which mention for example that "The accuracy of SMAC is within 2%–3%, if slope effects are mild, and high viewing and solar angles are avoided.". As for geometry, SMAC is found to be accurate for zenith angles up to 60-70° when the considered plane-parallel assumption is valid. For the selected stations, view zenith angle is always below this threshold, but solar zenith angle goes beyond each time we are close to sunrise or sunset. SMAC limitations in this case could result in an inferior quality of the SEVIRI-retrieved surface reflectance at the beginning and end of the day (see Fig. A2b), which after using it in the data simulation could degrade the realism of the synthetic data at these times of the day. However, it must be noted that these potential inaccuracies not only should affect a few data points, but also should not directly affect the inversion results since the SEVIRI-derived surface reflectance is the "true" surface reflectance in the synthetic data. Indeed, surface-related inversion errors are avoided since the same SEVIRI-derived surface reflectance is used for inversion. Only the potentially degraded

realism of synthetic data close to sunrise and sunset could impact our analyses conclusions, as it is now mentioned in the manuscript (see new sentence in L602: "We note the sole potential exception of high zenith angles for which the quality of the gas correction applied in step 1 may be impacted by the considered plane-parallel assumption."). Finally, it is worth noting one more time that the present study has not been carried out with TOA data but gas-free TOL simulations, and therefore the impact of SMAC biases on the found FCI performances is not relevant here.

- *Also, there are two Ceamanos et al 2023, so there might need to be an "a" and "b" discriminant depending on how this journal does it.* We thank the reviewer for spotting this issue, which we have corrected.

- *Line 595: Is there any random or systemic uncertainty that needs to be accounted for using this polynomial fit, in particular for the FCI 510nm and FCI 640/SEVIRI 635 channels, and, if so, how is it being accounted for? From Figure A1 it seems like there might be some systemic biases being introduced here.* The goal in our study is to simulate realistic FCI synthetic data, and that includes using surface reflectance as realistic as possible in the simulation. We honestly believe that the followed approach (described in Appendix A2) allows us to reach this goal. In particular, we use state-of-the-art satellite-derived data to spectrally extrapolate the 15-min surface reflectance obtained from SEVIRI VIS06 to the new FCI channels. While some uncertainty may be introduced by this approach (e.g., the polynomial fitting), we believe that the resulting surface reflectance is of high realism, and in particular its spectral dependence. Note in Fig. A2a that FCI points were obtained using the GRASP/POLDER spectral ratios only, and therefore are not expected to follow the fitting curve exactly. Again, it is important to note that the resulting surface reflectance becomes the "truth" in the synthetic data simulation and posterior inversion, and therefore residual biases such as the ones commented by the reviewer cannot easily impact the drawn conclusions. We added the following line on this topic in the revised manuscript (see L594): "It is worth noting that GRASP/POLDER data are only used to spectrally scale the SEVIRI-derived 15-min surface reflectance to impose a realistic spectral variation in synthetic data.".

- *Line 602-603: How bad of an assumption would a spectrally invariant BRDF shape be? Is there a quick thought or reference you can put in here to give an estimate on how it might impact the retrieval capabilities, particularly at larger angles?* Please refer to our previous answer, which also applies here. A spectrally invariant BRDF shape is considered to provide realistic enough FCI synthetic observations, and the same assumption is made in inversion. See new sentence in L601 mentioning this point.

- *Line 625: For clarification, this was used to estimate the surface reflectance in Figure A1b?* Yes. Please note that this figure (now Fig. A2b) has been updated, as the previous one corresponded to a wrong instrument channel.

- *639: view of both solar and viewing zenith angles for the mu terms?* You are right, but please note that this appendix was removed for brevity as suggested by other reviewers. Readers are now referred to Ceamanos et al, 2023 for details on the MSA RTM equations.

- *Line 660: To me, Figure C2 looks to have a pretty similar error, if not more in some day/variable combinations, as C1. Could these be better visualized as a scatter plot or a Tukey mean-difference plot?* Please note that the main visual differences between Fig. D1 and Fig. D2 (in the revised manuscript) happen for the "b" plots corresponding to Mongu Inn station, notably for the retrieved AOD time series. Table 6 confirms the better results of FLOTSAM in comparison to MSA (with respectively 4 and 1 "best" scores when they are both compared to the reference DOAD)

---

## Author Comment (AC3)

**EGUSPHERE-2025-1353**

-

**Response to the reviewers**

Adèle Georgeot and co-authors

June 5, 2025

**Reviewer 3**

*The paper presents a sensitivity study for the retrieval of AOD using FCI. The topic is important for the new sensor; however, the overall structure of the paper is difficult to follow. The authors divide the study into several sections based on different settings, channels, and methods, but the logic behind the sensitivity study is not clearly articulated. The authors are encouraged to reconsider the organization of the paper. For instance, the one-channel retrieval could be presented as a subsection alongside the two-channel retrieval, clearly indicating the added value of including more channels. Similarly, the distinction between the idealized cases and real-world cases should be better structured, with a clear conclusion on how ignoring real-world conditions affects retrieval accuracy.*

We thank the reviewer for the valuable comments and the time spent reviewing our work. We have attentively considered all the raised issues and have produced a revised version of the manuscript addressing most of them. Please find below our replies to the comments and questions from the reviewer (shown in italics). The line numbers given in our answers correspond to the revised manuscript, with the "track changes" mode on.

Regarding clarity, the manuscript has been restructured and shortened to make it more easy to follow. This was achieved by removing some parts of the initial manuscript (e.g., Figure 1, Sect. 4.3.1, equations in Appendix B) and moving some others to the appendices (e.g., Sect. 2.3, Sect. 4.1 and Sect 4.2). Also, many parts of the manuscript have been revised to improve the clarity of the goals that we pursue, the purpose of experiments and the main assumptions (e.g., the end of the introduction, Sect. 2.1 and the beginning of Sect. 3 and Sect. 4). Moreover, we have renamed sections 3 and 4 to "Retrieval of AOD from single channels" and "Joint retrieval of AOD and FMF using two channels" because the important distinction between the sections is that we retrieve AOD in one, and AOD+FMF in the second, and not that we use one channel or multiple ones. With all this, we believe that the revised manuscript enables to catch the main messages more easily now, while keeping the essential information.

Finally, the connection of the present work with real world conditions has been clarified in the article. In addition to the discussion in Sect. 3.2.1 and Sect. 3.2.2, we have added a new paragraph in the "Conclusions" section: see L493 "It must be noted that the retrieval accuracies observed in this study may be degraded when processing real FCI data, mainly due to differences between the optical properties of the aerosol models used for inversion and those of the real world aerosols observed by FCI (e.g., showing a large variability of mixtures in the case of desert dust and biomass burning particles). In the near future, the inversion methods used in this work will be tested with actual FCI data and real world limitations will be determined. For example, the consistency of the hybrid model used for FMF/AOD inversion with respect to the natural variability of smoke and dust particles will be assessed. Furthermore, the use of daily averages of retrieved AOD and FMF to constraint the second inversion of the two-step method will have to be circumvented if we are to meet near real time constraints."

*Some more specific comments are listed below:*

- *P155–P160: The radiative transfer model used for the simulations is critical. More detailed information about the strengths and limitations of the aerosol and surface simulations within the model is needed. For example, is the two-layer assumption sufficiently accurate? What are the potential impacts of the aerosol vertical distribution on the results?* We chose the DOAD radiative transfer model for our FCI data simulation due to its well-known high accuracy in remote sensing applications. DOAD consists in combining successive reflections, back and forth, of the transmission of the radiance through two layers to model multiple scattering (Lenoble, 1993), while taking into account light polarization (i.e., the geometrical orientation of the radiation waves oscillations) (de Haan et al., 1987). DOAD limitations include the use of a plane-parallel assumption, which is not correct for extreme geometries, and a low processing speed. We now describe DOAD in more detail with the following text that has been added to the revised manuscript (L124: "Simulations are made with the accurate doubling-adding (DOAD) radiative transfer model (de Haan, 1987), which models multiple scattering by combining successive reflections, back and forth, of the transmission of the radiance through two layers (Lenoble, 1993), while taking into account light polarization.". In our simulations, DOAD was run considering a standard atmosphere made of 30 layers, with aerosols being present in a few layers near the surface only. The mention of "two layers" in the original manuscript referred to the way DOAD solves the radiative transfer equation (and in particular the multiple scattering) between two "given" adjacent layers. For further understanding, the reviewer can take a look to the figure below (Fig. 1). It must be noted, however, that atmospheric vertical distribution is not very important in our simulation of FCI top-of-layer (TOL) reflectance because gases are not taken into account, and therefore the vertically-dependent gas-aerosol coupling is excluded from our simulations. This makes the potential impact of aerosol vertical distribution in our study very low. We remind the reviewer that the simulation of TOL reflectance (i.e., including aerosol and surface contribution, but excluding gas effects) and not TOA reflectance was made for the sake of simplicity and so that we could concentrate on aerosol-surface decoupling (i.e., the essential issue in aerosol remote sensing; see L127 "Synthetic data are simulated without including atmospheric gases for the sake of simplicity and to focus on the retrieval of aerosol properties"). Correction of FCI observations for gas contribution is being currently addressed in our team and will be published in a separate publication.

- *Using only two scenarios to represent real-world conditions is insufficient, particularly for assessing anthropogenic contributions. Additional cases representing various anthropogenic aerosol types are recommended.* We agree that knowing the FCI performances for "typical" aerosol conditions (e.g., continental aerosols with an anthropogenic influence) is also important, but we preferred to conduct our study on more extreme and therefore more retrieval-challenging aerosol conditions for which the previous SEVIRI instrument showed limitations. While Ceamanos et al. (2023) and Georgeot et al. (2024) showed a good agreement of SEVIRI AOD retrievals with reference data in regions with standard anthropogenic-influenced aerosol conditions (e.g., Europe, or South America when fire smoke is not present), they found much greater errors in bright surface regions such as Northwest Africa (corresponding to case study 1 in the present work) and in some areas of Southwest Africa due to aerosol-unfavorable geostationary satellite geometry (corresponding to case study 2). Naturally, AOD retrieval over more "aerosol-standard" regions (i.e., with an already acceptable AOD information content from SEVIRI) will improve with FCI thanks to the general increase in AOD sensitivity made possible by this new sensor. Finally, we agree that anthropogenic particles can also be found in extreme aerosol conditions such as in the case of very high pollution. Nonetheless, we decided to focus on dust and smoke events only for the sake of brevity (the manuscript is already quite long as it was noted by other reviewers) and because these events (i) are more frequent in the Meteosat's field of view (contrary to Himawari's, with frequent high pollution conditions in big cities such as Beijing) and (ii) are related to more challenging retrieval configurations (i.e., bright surfaces in the north African case study 1 and unfavorable geometry in south African case study 2). The following sentences on this subject has been included in Sect. 2.2 (L141): "These two case studies are chosen because of the high occurrence of desert dust and wildfire smoke events in the Meteosat's field of view and the SEVIRI retrieval limitations observed in these cases. Other challenging aerosol scenarios, such as heavy pollution episodes, have not been taken into account here for the sake of brevity and because they are less frequent in the Meteosat's field of view than in that of other geostationary missions

[Figure]

Figure 1: Principle of DOAD: We consider two layers of width d1 and d2. When the incident beam L crosses the first layer, part of it will be reflected and leave the layer (see R1 in the scheme) and part of it will be transmitted to the second layer. Then, the transmitted beam T1 will either be transmitted fully through the layer and leave the medium (T2), or will be reflected and therefore come into the first layer again (R3 ). In this first layer, the beam will either be reflected or transmitted, and so on. Basically, the part of the beam that changes direction is the reflected one, and the part that stays in the same direction is the transmitted one. In order to perform this method, we need to know the reflectance and transmission functions of the two layers.

(e.g., Himawari covering the big and often heavily polluted cities of East Asia)". Finally, the "Conclusions" section now reads (L480): "It is also worth noting that the FCI performances for AOD retrieval in less extreme and therefore less retrieval-challenging aerosol conditions than the ones considered in the selected case studies (e.g., anthropogenically influenced continental aerosols) are also expected to improve with respect to SEVIRI's."

- *Clarification is needed on how the AERONET-based aerosol models from MAIAC are implemented in the simulations. Specifically, which parameters are used as inputs? Models 6 and 7 are mentioned—are these fixed models, or are they updated using real-time AERONET data?* The MAIAC/MODIS C6 desert dust and biomass burning models (respectively, models 6 and 7) are introduced now in Appendix A1. These models are defined in Lyapustin et al. (2018) through their microphysical properties (i.e., particle size distribution, refractive index, sphericity). Mie and T-matrix codes included in the MOPSMAP software were used to calculate the optical properties (e.g., scattering phase function, single scattering albedo) of spherical smoke and non-spherical dust particles, respectively. The following sentence appears now in the revised manuscript (see L538): "Calculations are done taking into account the FCI spectral responses, and using a Mie code for spherical smoke particles in model 7 and a T-matrix code for non-spherical dust particles in model 6.". Finally, some clarification on the dynamism of these models was also added as it follows (see L534): "MAIAC aerosol models were built based on AERONET measurements that were fitted to by an AOD-dynamic bi-modal aerosol distribution. This means that size distribution, and therefore aerosol optical properties, vary with AOD, but remain constant with time.".

- *Regarding surface reflectance, the authors state that POLDER products were used and fitted to the FCI wavelengths. A thorough validation of this fitting procedure is necessary. Why not use a well-established BRDF model for the simulation instead?* As explained in Appendix A2, we use highly realistic surface reflectance that naturally varies every 15 minutes due to the combination of BRDF effects and the GEO-typical diurnally changing solar geometry. This

was made possible by estimating for each selected station the corresponding 15-min surface reflectance from real SEVIRI VIS06 observations. This was preferred to the fitting of a surface BRDF model, which will show differences with real BRDF as most well-established models are semi-empirical. GRASP/POLDER data were only used in a second time to derive the spectral ratios that are used to "scale" the SEVIRI-derived 15-min surface reflectance spectrally, from the SEVIRI VIS06 channel to each of the considered FCI channels. The goal here was to obtain a realistic spectral variation of surface reflectance (which is proved to be essential in aerosol satellite retrieval in our work). Therefore, the potential residual uncertainties coming from the GRASP/POLDER-based polynomial fit are not relevant as long as the resulting spectral surface reflectance captures well the natural spectral variation, which is proved to be the case. The final surface reflectance (including any residual uncertainty) is considered to be the "true surface" in our synthetic observations, and therefore cannot affect the inversion results obtained in our study. Finally, the synthetic data (and indirectly the considered surface reflectance) were satisfactorily validated as presented in the appendices (e.g., see Table A1 and Fig. A3). We added the following lines on this topic in the revised manuscript (see L594 and L601): "It is worth noting that GRASP/POLDER data are only used to spectrally scale the SEVIRI-derived 15-min surface reflectance to impose a realistic spectral variation in synthetic data" and "Despite these assumptions, the realism of surface reflectance obtained with the presented approach is judged to be high overall".

- *In the Gaussian noise section, is the wavelength-dependent behavior of noise considered? If so, what are the impacts of this dependence?* Indeed, we used wavelength-dependent SNR values (i.e., 25 for channels VIS04, VIS05 and NIR22 and 30 for channel VIS06 according to Holmlund et al. 2021) to define the Gaussian noise that was added to the synthetic data. This is explained in "Step 3" of Appendix A1 (see L559), and also in a new Table 1 that we included to improve the description of FCI channels. However, the relatively weak spectral change in SNR (n.b., SNR only differs for channel VIS06) does not result in a relevant impact on the synthetic data, nor their inversion, the main reason being that SNR is high for all channels.

- *Degrees of Freedom for Signal (DFS) is a useful theoretical metric for understanding information content, but I strongly recommend that the authors also evaluate retrieval uncertainty by comparison with real measurements—for example, comparing simulated and observed surface reflectance, TOA reflectance, and retrieved AOD.* The evaluations suggested by the reviewer already exist. First, synthetic data are validated in Appendix A3, in which the simulated time series of satellite TOL (and not TOA) reflectance is compared with real SEVIRI observations for all selected stations. The good agreement (mean RMSE of 0.016 and mean bias of 0.002) as it is shown in Table A1 and Fig. A3 proves not only the high quality of synthetic data, but also the realism of the inputs used for simulation (e.g., the surface reflectance). The direct validation of surface reflectance is not possible since there is not ground measurements of bi-directional reflectance at the selected stations, but in our opinion the indirect validation based on the successful evaluation of the FCI synthetic data is proof enough. Note that ground measurements would have been used in the simulations if they existed. Second, retrieved AOD (and FMF) is extensively validated (in Sect. 3.2, Sect. 4.2.2 and Sect. 4.3.2) through a comparison with the input AOD (and FMF, respectively), which in our case comes from AERONET measurements. It is important to remind here that our study was conducted based on synthetic data, mainly because in this way all the analyses made in our work can be precisely validated using the known inputs used for data simulation. For example, this helped us to know which individual FCI channel contains the highest AOD information content (through the DFS analyses) and which enables the most accurate retrievals. This would have not been possible with real FCI observations (which by the way were not available at the time of writing this manuscript) for which most parameters such as surface reflectance would have been unknown (or known with a certain degree of accuracy at best). The importance of synthetic data is reminded in the revised manuscript in L130 ("The choice to use synthetic data is made to precisely evaluate the retrieval performances with FCI, which would not be possible with real satellite observations for which many parameters are unknown.").

- *P218: How are the observation and prior covariance matrices incorporated into the DFS calculation? This needs further clarification.* The values of the observation and prior covariance

matrices used to compute DFS are given at the beginning of Sect. 3.1: "We calculate DFS following Eq. 2 for each station included in the FCI-like synthetic data setting Sa to 0.05 and $S_\epsilon$ to 0.0001 according to Georgeot et al. (2024), who found these values to provide accurate AOD retrievals from SEVIRI observations based on comprehensive experiments.". In fact, Georgeot et al. (2024) and Ceamanos et al. (2023) proved these values to be appropriate for AOD retrieval from Meteosat data based on the good results obtained after extensively processing real SEVIRI observations. We now explain the choice of these values in L215 for example: "For example, the value of $S_\epsilon$ was set to encompass the uncertainty of the SEVIRI channel VIS06 (equal to 3% according to Luffarelli et al. (2019)) and other errors operating in the inversion process".

- *The sensitivity study sections are largely qualitative. A more comprehensive and quantitative analysis is strongly recommended.* First, please note that we have renamed the original Sect. 3.1 and Sect. 4.3 (now Sect. 4.1) in which DFS are used to "Information Content Analysis" according to the suggestion of another reviewer. Second, we honestly believe that our DFS-based analyses are largely quantitative since we not only provide mean and case study-dependent statistics (e.g., see Table 3), but we also zoom in on the scores obtained for selected stations (e.g., Fig. 3 and Fig. 6). While we agree that conclusions drawn based on the DFS-based analysis only give a first idea of the FCI potential (indeed, DFS calculation is theoretical and may therefore be not 100% conclusive; is this what the reviewer meant when talking about "qualitative results"?), they are later confirmed by results obtained in subsequent sections, in which synthetic data are inverted and results are comprehensively and quantitatively assessed with the reference input data (see Sect. 3.2, Sect. 4.2.2 and Sect. 4.3.2). We added some clarification on this in the revised manuscript (see L211: "The goal of this step is to get a first idea of which channels contain the most information on aerosols and, in particular, on AOD, before processing the synthetic data for inversion", and L339: "to get a first idea of the FCI potential before processing the synthetic data for FMF/AOD inversion")

- *Section 3.2.1: In the ideal case, why does such a large retrieval error occur? The authors briefly mention potential sources such as noise and the inversion method, but this requires more detailed investigation and explanation.* Please note that the "Ideal Conditions" section title was misleading, and has been therefore modified to simply "Results" (see Sect. 3.2.1). Indeed, while input data are perfectly known in these experiments, data inversion is not made in ideal conditions since it can be impacted by several error sources. The main error source comes from the biases of the simplified radiative transfer model used for inversion (i.e., MSA, compatible with operational constraints), which is not the same than the one used for data simulation (i.e., DOAD, operational processing incompatible). The increase of MSA biases with AOD (see Fig. B1 in Appendix B) results in an increasingly large retrieval error on days with higher AOD (see June 23, 24 and 25 in Fig. 4a). Furthermore, inversion uncertainty can also come from the additive Gaussian noise and the Levenberg-Marquardt method, which may fail in finding a solution in some challenging inversion cases. All the existing biases can have an impact on the inverted AOD quality, with the magnitude of this impact potentially varying during the day because of the also diurnally varying AOD information content in GEO satellite observations. This variation comes from the diurnal change in solar geometry, which results in a diurnally varying aerosol scattering (peaking when SZA is high, and therefore scattering angle is low; eg. at the beginning and end of the day over regions along the 0° meridian) and an also diurnally varying surface reflectance (peaking when SZA is low, and therefore scattering angle is high; eg. around local noon along the 0° meridian, see TOL reflectance curves in Fig. 3a). As a result, AOD retrieval becomes more biased at noon over Saada for example (see the higher biases in the first four days of Fig. 4a, at noon). Additional explanations have been now included in L274 ("These uncertainties can result in retrieved AOD errors, which can vary in magnitude (e.g., due to the increase of MSA bias with AOD; see Fig. B1) and according to the time of day (i.e., because AOD information content varies during the day due to the also diurnally varying solar geometry in geostationary observations).").

- *Why is a one-channel retrieval necessary given that the sensor has multiple channels? Will both one-channel and two-channel retrievals be used in practice when the satellite becomes operational?* We agree with the reviewer that aerosol remote sensing can benefit from multi-channel retrieval. However, the goal of Sect. 3 is to quantify the retrieval potential of each channel separately, so

that the obtained results can be used in developing future aerosol retrieval algorithms for FCI data. The following text was included at the beginning of Sect. 3 (L201): "In this section, we analyze each channel separately due to two main reasons. First, previous works performed AOD retrieval from SEVIRI data using the channel VIS06 only (Ceamanos et al. (2023a), Georgeot et al. (2024)), and therefore the results obtained in this section can help to assess how much FCI can improve AOD estimation using a similar single-channel algorithm. Second, knowing the sensitivity to AOD of each channel can help identifying the most appropriate to be used in multi-channel retrieval algorithms, which have proved to benefit aerosol remote sensing (Lyapustin et al. (2018), Limbacher et al. (2024))". It must be noted that the results obtained in Sect. 3 helped us to select channels VIS04 and NIR22 for the AOD/FMF retrieval described in Sect. 4.

- *All similar comments regarding Section 3 also apply to Section 4.* Further explanations have been included in Section 4 according to the reviewer's comments on Section 3 wherever they applied. For instance, the beginning of Sect. 4 has been clarified, the previous Sect. 4.1 and 4.2 have been moved to Appendix, and Sect. 4.3.1 has been removed for the sake of brevity and clarity. Also, we have renamed the previously named "Sensitivity Study" section 4.1 as "Information Content Analysis", which is much clearer in our opinion. We have also given the values of the covariance matrices used for calculating DFS in this section (see L343: "The measurement covariance matrix $S_\epsilon$ was set as follows: $S_\epsilon = \begin{pmatrix} 0.0001 & 0.00052884 \\ 0.00052884 & 0.0001 \end{pmatrix}$. The variance (0.0001) for channels VIS04 and NIR22 was chosen accordingly to Sect. 3. The covariance values (0.00052884) were calculated by using the whole synthetic dataset and calculating the covariance between the TOL reflectance of these two channels. As for the a priori covariance matrix, it was set to $\begin{pmatrix} 0.2 & 0.0 \\ 0.0 & 0.5 \end{pmatrix}$ following a similar empirical approach as in Georgeot et al. (2024)."

---

## Author Response (AR2)

**EGUSPHERE-2025-1353**
**Response to the editor**

Adèle Georgeot and co-authors

August 12, 2025

**Note to the editor**

Thank you very much for your answer and your decision on our manuscript. We just uploaded all the required files, including the manuscript revised according to the production office suggestions. However, please note that, although we removed the placeholders "TEXT", we could not understand why some figures were misplaced, even after reading the manuscript composition suggestions. We hope that the production office will be able to place the figures correctly when editing the tex file.

Please note that after the first revision we had renamed the paper as "*Towards improved retrieval of aerosol properties from the **geostationary** orbit with the new Meteosat Third Generation-Imager **geostationary** satellite*". Since this title contained the word "*geostationary*" twice, we now suggest to remove the second occurrence so that it reads "*Towards improved retrieval of aerosol properties from the geostationary orbit with the new Meteosat Third Generation-Imager satellite*". We hope that you agree with this modification. We have updated the title in the tex and pdf documents that we uploaded accordingly.

Best regards,
Adèle Georgeot and co-authors.